# CrackInstSynth: Topology-Aware Generative Data-Augmentation Framework for Crack Instance Segmentation

## Abstract

Instance-level crack segmentation is critical for automated structural health monitoring of tunnels and bridges, yet progress is limited by the scarcity of densely annotated datasets with instance-level labels. To address this gap, we make two key contributions. First, we introduce CrackInst1K, to our knowledge the first publicly available instance-level crack segmentation dataset, comprising 1025 high-resolution tunnel images with pixel-accurate instance masks. Second, we propose CrackInstSynth, a generative data-augmentation framework that substantially enlarges instance-level crack corpora while preserving geometric and topological realism. CrackInstSynth integrates three coordinated modules: (i) Region-level Instance Placement (RIP), which partitions the canvas into quadrants to strategically position crack instances for diverse layouts; (ii) a Physics-driven Skeleton Generator (PSG) that enriches morphological variability by growing crack skeletons via physical simulation; and (iii) a Topology-Preserving Generation Module (TPGM) that employs a two-stage conditional diffusion pipeline (skeleton→mask, mask→image) to produce paired, width-aware instance masks and corresponding images while enforcing intra-instance topology and inter-instance separation. Extensive experiments show that augmenting real data with CrackInstSynth consistently improves the performance of multiple instance segmentation models on CrackInst1K and other benchmarks, validating both visual fidelity and downstream effectiveness. We will release CrackInst1K and CrackInstSynth to facilitate future research in structural health monitoring.

## 1 Introduction

Cracks on the surfaces of tunnels Lei et al. (2024c); Wang et al. (2023), bridges Lei et al. (2024c); Chen et al. (2022), pavements Lei et al. (2023; 2024a); Chen et al. (2025), and other civil infrastructure are critical indicators of material fatigue, water ingress, and progressive structural deterioration Yuan et al. (2024). Early and accurate detection enables preventive maintenance before minor defects escalate into serious hazards, thereby extending service life and reducing life-cycle costs Panella et al. (2022). For tunnel linings in particular, unchecked crack growth can quickly compromise structural integrity, whereas timely repair markedly reduces subsequent expenditures Wang et al. (2023).

While binary crack maps suffice for coarse assessment, downstream tasks—such as measuring per-crack size, prioritizing repair schedules, and tracking temporal evolution—require distinguishing individual cracks Zhao et al. (2024a); Lei et al. (2024c). Instance-level segmentation simultaneously localizes and uniquely labels each crack, yielding pixel-accurate masks that support geometric analysis, path tracing, and longitudinal studies. Compared with semantic segmentation, which merges all crack pixels into a single class, or detection pipelines that output only bounding boxes, instance segmentation provides the level of detail needed for automated, priority-driven maintenance strategies Shi et al. (2021). Fig. 1 illustrates a workflow for measuring tunnel-lining crack properties using instance segmentation.

Despite recent progress, crack segmentation research remains severely data-constrained. Public benchmarks such as CFD Shi et al. (2016), DeepCrack Zou et al. (2018), and CRACK500 Shi

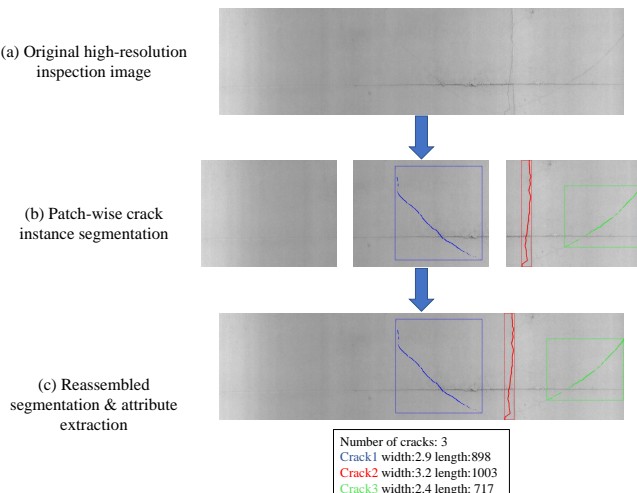

Figure 1: Workflow for instance-level crack segmentation and measurement. (a) Original high-resolution inspection image. (b) Patch-wise instance segmentation: the image is tiled into patches, and each patch is processed by an instance segmentation model to produce individual crack masks (blue, red, and green). (c) Reassembled segmentation and attribute extraction: patch-wise predictions are mapped back to the original image coordinates and merged to obtain image-level crack instances, from which geometric attributes (e.g., width and length) are computed and summarized.

et al. (2016) contain only a few hundred images, and collecting additional data is costly: a single tunnel survey typically yields fewer than 500 usable frames, while pixel-level annotation by safety experts can take several minutes per image Panella et al. (2022); Liu et al. (2019); Rezaie et al. (2020). The shortage is even more acute for instance-level labels: no open dataset currently provides pixel-accurate instance IDs (unique labels for each crack), and most studies rely on binary masks or sparsely annotated subsets Zhao et al. (2024a). This gap hampers automatic structural health monitoring and highlights the need for cost-effective methods to enlarge corpora for crack instance segmentation.

To mitigate the twin shortages of scale and instance granularity, we curate **CrackInst1K**, a publicly available tunnel-lining crack dataset comprising 1025 high-resolution images (1024×1024 pixels). Images collected across multiple years and tunnels are first tiled into patches and then randomly mosaicked to ensure de-identification while guaranteeing that each image contains at least one annotated crack instance. At a scale where ∼1,000 pixels correspond to ∼1,m on the lining, each image covers roughly 1,m$^2$. Compared with prior crack segmentation benchmarks, CrackInst1K provides explicit instance labels together with accurate geometric scale and focuses on two crack types that require precise sizing in tunnel and bridge structural health monitoring—hairline cracks and branched/intersecting cracks—while excluding map cracking (crazing), generally does not require exact dimensional measurement. This establishes a fine-grained reference set for geometry-accurate, instance-level evaluation.

Building on this resource, we develop **CrackInstSynth**, a topology-aware generative augmentation framework that produces diverse crack instance segmentation image–mask pairs without additional manual labeling. The pipeline first performs **Region-level Instance Placement (RIP)** by selecting one to three seed instances from CrackInst1K and placing their masks at random positions within a randomly selected canvas quadrant. Next, a **Physics-driven Skeleton Generator (PSG)** stochastically grows each seed's skeleton under a physics-based crack growth simulation, thereby increasing the informationtiveness of seeds. The grown seeds are then processed by a **Topology-Preserving Generation Module (TPGM)** with two diffusion stages: (i) a skeleton→mask stage, in which a pixel-space generative model conditionally produces width-aware instance masks; and (ii) a mask→image stage, in which a topology-consistent, ControlNet-style (TC-ControlNet) diffusion model, conditioned on the width-aware masks and a predefined text prompt, renders photorealistic crack images whose topology exactly matches the conditioning masks. By integrating these three

modules, CrackInstSynth generates diverse, label-aligned samples that augment real data and enable large-scale training of crack instance segmentation models.

We assess CrackInstSynth on CrackInst1K as well as the public DeepCrack Liu et al. (2019) and CRACK500 Shi et al. (2016) benchmarks. The evaluation covers both the visual fidelity of the synthetic images and the performance gains of the synthetic image-mask pairs provide when training state-of-the-art instance segmentation models. In every case, the augmented data yields clear improvements over baseline training sets, confirming the practical value of the proposed framework for crack analysis. We will release CrackInst1K and CrackInstSynth to support future research.

In summary, this paper makes the following contributions:

1. **CrackInst1K**: a publicly available dataset of 1025 tunnel-lining images ($1024\times1024$) with pixel-accurate per-instance masks. Images are de-identified via tile–mosaic preprocessing; the set focuses on hairline and branched/intersecting cracks, forming a fine-grained benchmark.

2. **CrackInstSynth**: a topology-aware generative augmentation framework that produces diverse, label-aligned image–mask pairs without extra labeling, integrating RIP, PSG, and TPGM with a two-stage diffusion pipeline (skeleton→mask, mask→image).

## 2 RELATED WORK

### 2.1 CRACK INSTANCE SEGMENTATION

Most deep learning studies on cracks still treat the problem as binary or semantic segmentation, classifying all crack pixels as a single class. Transformer-based detectors such as CrackFormer Liu et al. (2021) improve hairline preservation by modeling long-range context, yet intersecting cracks often remain merged, which limits geometric analysis that requires per-crack masks.

More recent work adapts generic instance frameworks to the thin-object regime of cracks. Mask R-CNN He et al. (2017) pipelines augmented with morphological closing reconnect fragmented masks and improve accuracy on tunnel linings Huang et al. (2022). Orientation-aware detectors represent a curved crack as a sequence of rotated segments to separate crossings and branches Chen et al. (2023). Lightweight one-stage networks enhanced with multi-head and triplet attention achieve real-time instance segmentation while boosting recall for fine structures Yu et al. (2025). Other approaches advance per-crack labeling but still struggle with continuity and ambiguous boundaries Zhao et al. (2024b); Lei et al. (2024c).

Topology-preserving techniques address these weaknesses. A differentiable connectivity loss penalizes broken masks Pantoja-Rosero et al. (2022), and ambiguity-aware representation learning refines uncertain crack edges Chen et al. (2024). Large benchmarks such as OmniCrack30k Benz & Rodehorst (2024) expand training data, yet none provide dense instance labels with calibrated metric scale. CrackInst1K and the CrackInstSynth framework fill this gap by supplying precisely scaled instance annotations and generating additional topology-consistent data, enabling large-scale instance segmentation without additional manual labeling.

### 2.2 GENERATIVE DATA AUGMENTATION

Conditional GANs were the first practical engines for paired data synthesis. pix2pixHD and SPADE translate coarse semantic masks into photorealistic surfaces and have been adapted to crack and road-defect imagery Wang et al. (2018); Park et al. (2019). Such GAN-based pipelines boost realism but offer limited geometric diversity and often break thin structures, restricting their value for topology-sensitive tasks.

Recent augmentation studies explore alternative generative cues. Cut-and-paste strategies like Insta-Boost warp foreground masks to create new layouts Fang et al. (2019), whereas domain-randomized renderers synthesize cracks by overlaying procedural textures on material maps Yang et al. (2020). MosaicFusion shows that a single diffusion pass can populate disjoint canvas regions with multiple labeled objects for detection and segmentation Xie et al. (2025), and Panoptic Diffusion embeds instance IDs in the latent space to reduce label misalignment Bansal et al. (2023). Although these

methods enlarge datasets efficiently, none explicitly maintain the connectivity and mutual separation required by elongated, intersecting cracks.

Latent diffusion models inject text or mask guidance into the denoising process, delivering higher fidelity and broader mode coverage Rombach et al. (2022). ControlNet Zhang et al. (2023), T2I-Adapter Mou et al. (2024), and related variants refine mask conditioning but still downsample masks and weaken structural cues. CrackInstSynth advances this line by coupling physics-driven skeleton growth with a two-stage, topology-preserving diffusion module, generating large volumes of geometry-faithful image–mask pairs tailored to crack instance segmentation.

# 3 METHODOLOGY

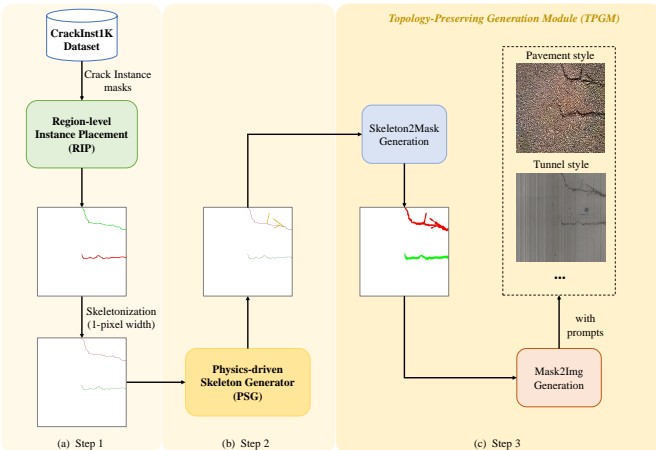

Figure 2: Overall workflow of the proposed CrackInstSynth framework. The pipeline comprises three sequential stages: (a) Region-level Instance Placement (RIP), (b) Physics-driven Skeleton Generator (PSG), and (c) Topology-Preserving Generation Module (TPGM), which performs Skeleton-to-Mask and Mask-to-Image synthesis while enforcing crack topology.

CrackInstSynth tackles the scarcity of crack-instance data by generating topology-consistent image–mask pairs in three stages, as shown in Fig. 2. **(a) Region-level Instance Placement (RIP)** selects one to three seed instances (masks) from the curated *CrackInst1K* dataset and placing their masks at random positions within a randomly selected canvas quadrant, producing diverse multi-instance layouts; the placed masks are then skeletonized to one-pixel width to prepare for physics-based growth. **(b) Physics-driven Skeleton Generator (PSG)** takes the one-pixel skeletons and stochastically grows each under a physics-based crack growth simulation, injecting physically plausible branching and increasing the informationtiveness. **(c) Topology-Preserving Generation Module (TPGM)** then runs a two-stage diffusion process: a *Skeleton2Mask* network inflates each skeleton into a width-aware instance mask, and a *Mask2Img* network—conditioning a topology-consistent, ControlNet-style diffusion model (TC-ControlNet) on the width-aware masks and style prompts (e.g., *pavement*, *tunnel*)—renders photorealistic crack images whose geometry and topology exactly match the conditioning masks.

## 3.1 REGION–LEVEL INSTANCE PLACEMENT

Let $\mathcal{D} = \{(\mathbf{M}_j, \mathbf{b}_j)\}$ denote the set of pixel masks $\mathbf{M}_j \subset [0, 1]^{H_0 \times W_0}$ in *CrackInst1K* and their axis–aligned bounding boxes $\mathbf{b}_j = [x_j, y_j, w_j, h_j]$. Given a blank canvas $\mathcal{C} \in \mathbb{R}^{H \times W \times 3}$ of size $H = W = 1024$, RIP synthesises a layout with $n \sim \mathcal{U}\{1, 2, 3\}$ instances.

The canvas is partitioned into four non–overlapping quadrants $\mathcal{R} = \{R^1, R^2, R^3, R^4\}$, each of size $512 \times 512$. RIP selects $n$ distinct regions $\{R^{\pi(1)}, \ldots, R^{\pi(n)}\}$ by a random permutation $\pi$, then places the $n$ sampled masks after an i.i.d. translation

$$\mathbf{t}_i \sim \mathcal{U}\Big([0,\, w_i^{\max}] \times [0,\, h_i^{\max}]\Big),$$
$$w_i^{\max} = w_{R^{\pi(i)}} - w_j, \tag{1}$$
$$h_i^{\max} = h_{R^{\pi(i)}} - h_j.$$

so that every translated box $\tilde{\mathbf{b}}_i = \mathbf{b}_j + \mathbf{t}_i$ is fully contained in its region. The procedure returns a colour canvas $\mathbf{I}_{\mathrm{RIP}}$, that is, a segmentation of multiple crack instances and the translated annotations $\{\tilde{\mathbf{M}}_i, \tilde{\mathbf{b}}_i\}$.

RIP can be summarized as follows: (i) we evenly divide a $1024 \times 1024$ canvas into four non-overlapping $512 \times 512$ regions, inspired by MosaicFusion Xie et al. (2025), to increase the information density per image; (ii) we then iteratively place up to three sampled crack instances at random into the four regions. This design encodes a civil-engineering prior: since *CrackInst1K* is calibrated such that $\sim 1000$,px $\approx 1$,m, a $1024 \times 1024$ canvas (about $1$, m$^2$) should typically contain no more than three cracks.

Unlike MosaicFusion, in RIP the instances are sampled at the $1024 \times 1024$ scale and then placed into $512 \times 512$ regions. We allow region overflow at placement time: portions extending beyond the assigned region are preserved, whereas any content outside the outer $1024 \times 1024$ canvas is clipped. This relaxation increases layout diversity and retains potential cross-region interactions (e.g., intersecting cracks), while keeping the global canvas consistent.

## 3.2 Physics-driven Skeleton Generator

For each seed mask $\tilde{\mathbf{M}}_i$ produced by RIP we extract a one-pixel-wide skeleton $\mathbf{S}_i = \mathrm{THIN}(\tilde{\mathbf{M}}_i)$ using Zhang–Suen thinningLam et al. (1992). Let $\mathcal{B}_i$ denote the axis-aligned bounding box of $\tilde{\mathbf{M}}_i$. PSG expands $\mathcal{B}_i$ by a scale factor $\alpha \in [1.2, 1.6]$ to obtain a *growth window* $\hat{\mathcal{B}}_i$ inside which a stochastic crack propagation process is simulated. Starting from $k$ randomly sampled pixels on $\mathbf{S}_i$ ($k \sim \mathcal{U}\{1, \ldots, M_k\}$), default $M_k = 4$, a random walk Lei et al. (2024a) adds new skeleton points until a maximum relative length $m = 0.8$ of the window is reached or a step limit is met:

$$\mathbf{S}_i^{\mathrm{new}} = \mathrm{RANDOMWALK}\big(\mathbf{S}_i,\, \hat{\mathcal{B}}_i,\, k,\, m,\, \mathit{max}, \ell, \theta\big),$$

where *max* are the maximum step counts, $\ell$ is the step length in pixels (default 2) and $\theta$ bounds the turning angle ($\pm 30°$). The walk is confined to $\hat{\mathcal{B}}_i$ to avoid inter-instance overlap.

See Apendix A.3.1 for the Algorithm 1 of RANDOMWALK.

Physically, cracks in the infrastructure structures propagate along principal stress directions while exhibiting stochastic branching. Embedding this behaviour via bounded random walks injects *plausible curvature, length variation and side branches*, thereby expanding the geometric distribution beyond the limited shapes in CrackInst1K.

## 3.3 Topology-Preserving Generation Module (TPGM)

TPGM takes as input the PSG-augmented skeletons. Let $\{\mathbf{S}_i^{\mathrm{new}}\}_{i=1}^{n}$ be the grown skeletons from PSG and define the *instance-ID skeleton map* $\mathbf{S}_{\mathrm{PSG}} \in \{0, \ldots, n\}^{H \times W}$ by assigning value $i$ to pixels on $\mathbf{S}_i^{\mathrm{new}}$ and 0 to background. Each pixel of $\mathbf{S}_{\mathrm{PSG}}$ encodes an *instance identifier* (0 for background, $1{:}n$ for cracks) along one-pixel-wide centerlines. TPGM produces a topology-aligned, *width-aware instance map* $\mathbf{M}_{\mathrm{WA}} \in \{0, \ldots, n\}^{H \times W}$ that expands each skeleton to its physical width, and a photorealistic image $\mathbf{I}$, through two diffusion stages.

Formally, TPGM learns

$$\mathcal{F}: \mathbf{S}_{\mathrm{PSG}} \longmapsto (\mathbf{M}_{\mathrm{WA}}, \mathbf{I}). \tag{2}$$

The mapping is realized by (i) a *Skeleton→Mask* diffusion network that inflates $\mathbf{S}_{\mathrm{PSG}}$ into $\mathbf{M}_{\mathrm{WA}}$; and (ii) a *Mask→Image* diffusion network that renders $\mathbf{I}$ conditioned on $\mathbf{M}_{\mathrm{WA}}$ while honoring the above topological requirements.

### 3.3.1 STAGE 1: SKELETON→MASK DIFFUSION

Given the one-pixel-wide instance-ID map $\mathbf{S}_{\text{PSG}} \in \{0, \ldots, n\}^{H \times W}$, the goal is to infer a width-aware instance map $\mathbf{M}_{\text{WA}} \in \{0, \ldots, n\}^{H \times W}$ that satisfies connectivity and separation. We adopt the *pixel-level Semantic Diffusion Model (SDM)* framework Wang et al. (2022); Lei et al. (2024a), conditioning directly on $\mathbf{S}_{\text{PSG}}$. Skeleton conditioning is injected via SPADE Park et al. (2019) layers placed in every upsampling block of the UNet; no additional modalities are required.

Pixel-space diffusion preserves high-frequency details and has been shown to outperform latent-space diffusion (e.g., LDM) on tasks requiring strict structural fidelity, such as medical shape synthesis Konz et al. (2024) and curvilinear object augmentation Lei et al. (2024b). Because Skeleton→Mask uses only a single conditioning map without text prompts, the pixel-level SDM is more efficient and easier to train than latent counterparts while retaining full spatial resolution.

### 3.3.2 STAGE 2: MASK→IMAGE DIFFUSION

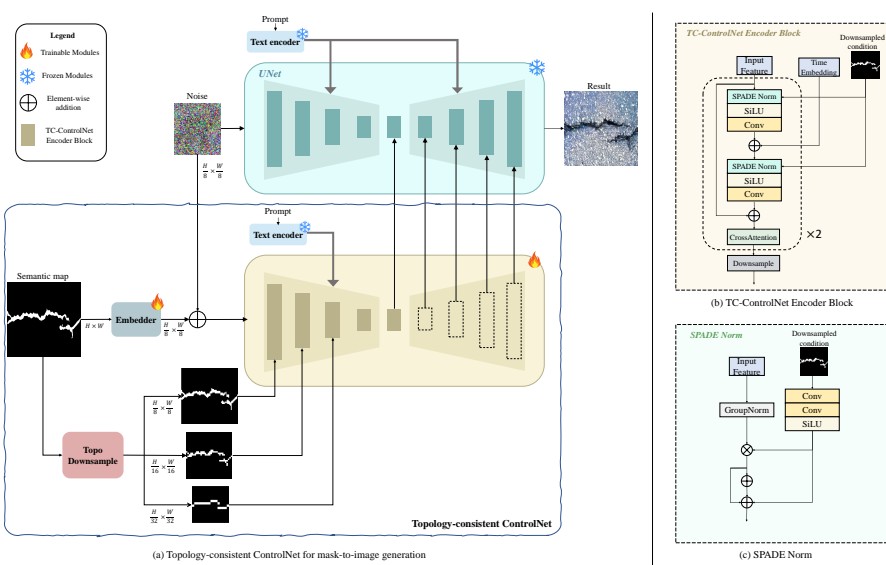

(a) Topology-consistent ControlNet for mask-to-image generation

(b) TC-ControlNet Encoder Block

(c) SPADE Norm

Figure 3: Topology-Consistent ControlNet (TC-ControlNet) for mask-to-image generation. (a) The upper path (blue) is the frozen Stable Diffusion UNet; the lower path (yellow) is the trainable ControlNet branch that injects multi-scale, topology-preserving mask features. A *TopoDownsample* module provides three scale masks whose region-adjacency graph is unchanged. (b) Encoder block details: the downsampled mask modulates features. (c) SPADE-Norm restores spatial cues "washed out" by standard normalization.

Latent diffusion models such as ControlNet Zhang et al. (2023) and T2I-Adapter Mou et al. (2024) often break fine connectivity when synthesizing from semantic maps: (i) the raw mask is down-sampled by a factor of eight via convolution/interpolation, destroying small-scale connectivity and topology; and (ii) subsequent normalization (e.g., GroupNorm) spatially averages feature statistics, further washing out geometry Park et al. (2019); Lei et al. (2024b). We therefore introduce a **Topology-Consistent ControlNet (TC-ControlNet)** that preserves crack connectivity and topology at both the input and feature levels (Fig. 3(a)). The two key adaptations relative to vanilla ControlNet are (1) a *TopoDownsample* module and (2) topology-aware feature modulation via SPADE Norm.

**(1) TopoDownsample module.** Before the mask enters the latent UNet it must be reduced to $\frac{1}{8}$, $\frac{1}{16}$, and $\frac{1}{32}$ of its original size. Naïve interpolation removes hairline cracks or merges adjacent regions. *TopoDownsample* performs this reduction while *exactly* preserving the connectivity and hole structure of every instance by casting downsampling as a small mixed-integer program (MIP) that assigns a component label to each low-resolution pixel. We maximize similarity to the original

mask while enforcing topology:

$$\max_x \sum_{m,i,j} w_m \, S_m(i,j) \, x_{m,i,j}$$

$$\text{s.t. } \sum_m x_{m,i,j} = 1 \quad \text{(exclusivity)}$$

$$\sum_{(i,j)\in\mathcal{R}_m} x_{m,i,j} \geq 1 \quad \text{(component survival)}$$

$$x_{m,i,j} + x_{m,i+1,j+1} \leq 1 \quad \text{(avoid diagonal bridges)}$$

Here $x_{m,i,j} \in \{0,1\}$ indicates whether low-res pixel $(i,j)$ is assigned to component $m$; $w_m$ weights components (foreground > background); and

$$S_m(i,j) = \underbrace{\frac{|m \cap \mathcal{N}(i,j)|}{|\mathcal{N}(i,j)|}}_{\text{local coverage}} + \lambda \underbrace{\frac{1}{1 + \min_{p\in\partial m}\|p - (i,j)\|}}_{\text{boundary proximity}},$$

with $\mathcal{N}(i,j)$ a circular neighborhood (radius $2s_k$), $\partial m$ the boundary of $m$, and $\lambda{=}0.5$. The first two constraints ensure every pixel takes exactly one label and no connected component disappears; the third keeps the background 4/8-connected at low resolution to prevent spurious holes. A small set of additional linear constraints (omitted here for brevity; see Appendix A.5) ensures boundary continuity so that foreground–background interfaces form a single closed loop. As a result, the downsampled masks preserve Euler characteristic, Betti numbers, and the region adjacency graph (RAG) of the original. Applying the MIP at three scales yields $\mathbf{M}^{(8)}, \mathbf{M}^{(16)}, \mathbf{M}^{(32)}$ fed to TC-ControlNet.

**(2) Topology-aware feature modulation via SPADE Norm.** To prevent normalization from blurring crack structure, every encoder block in the ControlNet branch replaces GroupNorm with SPADE Norm conditioned on topology-consistency masks (Fig. 3(b,c)). For an input feature tensor $f$ and a mask $\mathbf{M}^{(s)}$ at scale $s \in \{8, 16, 32\}$, the layer computes

$$\text{SPADE}(f, \mathbf{M}^{(s)}) = \gamma_s(\mathbf{M}^{(s)}) \, \tfrac{f - \mu(f)}{\sigma(f)} \, + \, \beta_s(\mathbf{M}^{(s)}), \tag{3}$$

where $\mu(\cdot)$ and $\sigma(\cdot)$ are per-channel statistics, and the spatially varying scale and shift maps $\gamma_s, \beta_s$ are produced by two $3{\times}3$ Conv–SiLU blocks. Feeding masks at three resolutions aligns the UNet's receptive field with expected crack widths and re-injects precise geometry that would otherwise be lost.

TC-ControlNet is trained with the same noise-prediction objective as vanilla ControlNet; only the mask embedder (some Conv layers) and the ControlNet branch are updated, while the Stable Diffusion backbone and the text encoder remain frozen.

## 4 EXPERIMENTS

### 4.1 EXPERIMENTAL SETUP

**Datasets.** All experiments are conducted on three high–quality crack datasets: *CrackInst1K*, *CRACK500* Shi et al. (2016), and *DeepCrack* Liu et al. (2019).

**Evaluation protocol.** Effectiveness is assessed from two complementary angles:

1. *Visual realism and consistency.* For every dataset we synthesise a set of image–mask pairs with CrackInstSynth+TC-ControlNet and compute
   - FID on RGB images (realism);
   - mIoU and absolute Betti errors $\beta_0$, $\beta_1$ between ground-truth masks and predictions of a pre-trained robust U-Net Lei et al. (2024b) (consistency).

2. *Downstream instance segmentation.* Each training set is enlarged five-fold using the full CrackInstSynth pipeline (only replaces TC-ControlNet with other generative models). A

standard Mask R-CNN He et al. (2017) detector is trained from scratch for $100$ epochs (batch size $\geq 8$) on the augmented data, and evaluated with $\text{mAP}_{50}^{\text{bbox}}$ and $\text{mAP}_{50}^{\text{seg}}$ on the held-out test split. For the semantic segmentation datasets *CRACK500* and *DeepCrack*, we consider there to be only one instance.

## 4.2 EXPERIMENT RESULTS AND DISCUSSION

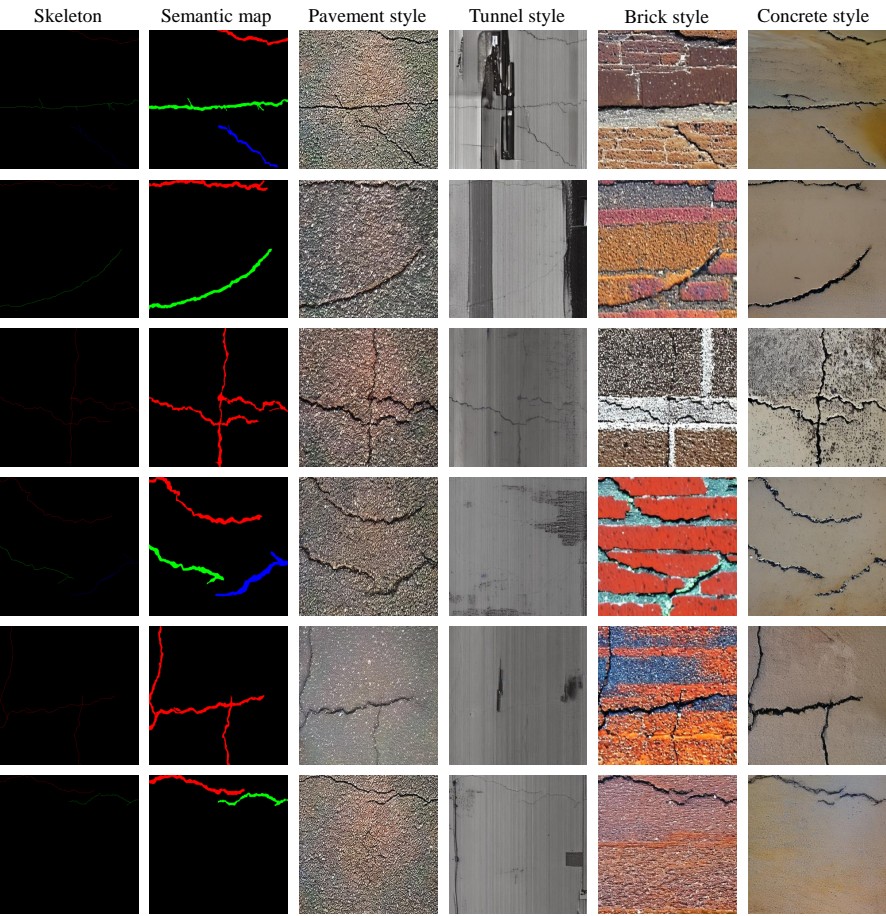

Figure 4: Qualitative results produced by **CrackInstSynth**. Columns (left→right): *Skeleton*, *Semantic map*, *Pavement style*, *Tunnel style*, *Brick style*, *Concrete style*. For each row we keep the mask (instance IDs shown by colours) fixed and only change the text prompt; TC-ControlNet transfers the background/material appearance while preserving per-instance topology and width, demonstrating label-faithful, multi-style rendering. Best viewed in colour and with zoom.

### 4.2.1 EVALUATION OF VISUAL REALISM AND CONSISTENCY

We compare TC-ControlNet with representative GANs (Pix2PixHD Wang et al. (2018), SPADE Park et al. (2019)), a pixel-space diffuser (SDM) Wang et al. (2022), and leading latent-space methods (T2i-Adapter Mou et al. (2024), FreestyleNet Xue et al. (2023), ControlNet Zhang et al. (2023), PLACE Lv et al. (2024), SCP-ControlNet Lei et al. (2024b)).

Table 1 shows that TC-ControlNet achieves the lowest FID and Betti errors and the highest mIoU on all three datasets. Relative to the strongest baseline (SCP-ControlNet) it cuts $\beta_0$ by 22%–30%, reduces $\beta_1$ by about $10\%$, and improves FID by 2–6, confirming that the TopoDownsample and SPADE-Norm mechanisms jointly preserve fine connectivity without sacrificing photorealism.

Table 1: Visual realism and topology consistency on CRACK500, Deepcrack, and CrackInst1K. Best results are in **bold**.

| Datasets | CRACK500 | | | | Deepcrack | | | | CrackInst1K | | | |
|---|---|---|---|---|---|---|---|---|---|---|---|---|
| Methods | mIoU ($\uparrow$) | FID ($\downarrow$) | $\beta_0$ ($\downarrow$) | $\beta_1$ ($\downarrow$) | mIoU ($\uparrow$) | FID ($\downarrow$) | $\beta_0$ ($\downarrow$) | $\beta_1$ ($\downarrow$) | mIoU ($\uparrow$) | FID ($\downarrow$) | $\beta_0$ ($\downarrow$) | $\beta_1$ ($\downarrow$) |
| pix2pixHD | 46.7 | 118.4 | 0.171 | 0.0120 | 48.2 | 148.9 | 55.00 | 37.45 | 54.2 | 142.3 | 52.04 | 32.44 |
| SPADE | 64.3 | 100.8 | 0.164 | 0.0115 | 63.4 | 137.5 | 50.84 | 33.35 | 74.0 | 125.4 | 39.21 | 26.15 |
| SDM | 62.1 | 98.3 | 0.140 | 0.0091 | 62.1 | 163.1 | 52.71 | 31.58 | 73.1 | 126.0 | 40.14 | 28.47 |
| T2i-Adapter | 52.4 | 94.5 | 0.151 | 0.0070 | 64.2 | 136.5 | 53.33 | 31.61 | 72.2 | 129.5 | 42.24 | 30.51 |
| FreestyleNet | 66.7 | 89.7 | 0.161 | 0.0088 | 65.3 | 122.1 | 48.51 | 31.46 | 74.8 | 113.1 | 36.14 | 26.86 |
| ControlNet | 73.4 | 90.3 | 0.168 | 0.0090 | 68.3 | 123.5 | 49.45 | 29.94 | 76.7 | 110.3 | 36.80 | 26.31 |
| PLACE | 71.3 | 88.6 | 0.124 | 0.0085 | 68.7 | 118.7 | 46.14 | 30.21 | 76.6 | 105.4 | 35.47 | 24.89 |
| SCP-ControlNet | 73.9 | 85.4 | 0.091 | 0.0073 | 67.6 | 117.6 | 44.77 | 31.22 | 76.0 | 103.8 | 34.28 | 22.95 |
| TC-ControlNet (ours) | **75.2** | **82.3** | **0.071** | **0.0050** | **70.9** | **111.8** | **41.05** | **25.31** | **79.7** | **101.9** | **32.35** | **20.09** |

Table 2: Downstream segmentation performance (Mask R-CNN) after $5\times$ data augmentation on Deepcrack, CRACK500 and CrackInst1K. Best results are in **bold**.

| Datasets | CRACK500 | | Deepcrack | | CrackInst1K | |
|---|---|---|---|---|---|---|
| Methods | mAP$_{50}^{bbox}$($\uparrow$) | mAP$_{50}^{seg}$($\uparrow$) | mAP$_{50}^{bbox}$($\uparrow$) | mAP$_{50}^{seg}$($\uparrow$) | mAP$_{50}^{bbox}$($\uparrow$) | mAP$_{50}^{seg}$($\uparrow$) |
| Original | 85.1 | 75.4 | 86.6 | 84.6 | 84.2 | 70.1 |
| pix2pixHD | 86.3 | 77.8 | 87.4 | 87.1 | 86.3 | 73.8 |
| SPADE | 87.9 | 79.6 | 88.9 | 88.0 | 88.1 | 76.5 |
| SDM | 87.4 | 79.1 | 88.5 | 87.6 | 87.6 | 76.0 |
| T2i-Adapter | 87.0 | 78.6 | 87.2 | 88.5 | 87.1 | 75.4 |
| FreestyleNet | 87.6 | 80.5 | 88.1 | 89.4 | 89.3 | 77.8 |
| ControlNet | 87.5 | 81.1 | 88.2 | 90.5 | 88.6 | 80.0 |
| PLACE | 87.0 | 81.6 | 88.8 | 91.0 | 89.4 | 79.2 |
| SCP-ControlNet | 87.4 | 82.5 | 89.0 | 90.9 | 89.1 | 81.1 |
| TC-ControlNet | **88.7** | **84.2** | **89.7** | **92.2** | **91.2** | **83.3** |

### 4.2.2 EVALUATION OF DOWNSTREAM SEGMENTATION PERFORMANCE

Table 2 reports Mask R-CNN performance after augmenting each training set to five times its original size with different generators. CrackInstSynth paired with TC-ControlNet yields the highest mAP$_{50}^{bbox}$ and mAP$_{50}^{seg}$ on all three benchmarks, surpassing the strongest baseline (SCP-ControlNet) by up to $+1.5$ bbox AP and $+2.1$ mask AP. The gains over the "Original" rows confirm that the synthetic imagery is not only realistic but also *task-useful*, boosting instance detection and segmentation accuracy without additional manual labels.

### 4.2.3 VISUALIZATION AND QUALITATIVE ANALYSIS

Fig. 4 provides visual evidence that the proposed pipeline preserves crack geometry while offering flexible appearance control. TC-ControlNet renders photorealistic textures in various distinct styles, coarse asphalt, ribbed concrete tunnel brick masonry facade and concrete structural surface.

Table 3: Ablation study on **CrackInst1K** using downstream instance-segmentation metrics. Each variant removes or alters one component of CrackInstSynth.

| ID | Method | mAP$_{50}^{bbox}$ ($\uparrow$) | mAP$_{50}^{seg}$ ($\uparrow$) |
|---|---|---|---|
| **A0** | Original training set (no aug.) | 84.2 | 70.1 |
| **A1** | No Region-level Placement (naïve paste) | 86.4 | 75.4 |
| **A2** | No Physics-driven Skeleton Growth | 86.6 | 76.3 |
| **A3** | Vanilla ControlNet (no topology branch) | 88.6 | 80.0 |
| **A4** | TC-ControlNet w/ TopoDownsample only | 86.4 | 76.5 |
| | TC-ControlNet w/ SPADE only | 89.1 | 81.1 |
| **A5** | **Full CrackInstSynth (ours)** | **91.2** | **83.3** |

### 4.2.4 ABLATION STUDY

Table 3 isolates the contribution of each novel component on **CrackInst1K** dataset, reported with Mask R-CNN He et al. (2017) mAP after $5\times$ augmentation:

**A1** *No Region-level Placement.* Replacing RIP with naïve cut–paste lowers mask AP by $7.9$. Overlapping or truncated instances therefore hurt detector training even when image realism is preserved. **A2** *No Physics-driven Skeleton Growth.* Skipping PSG removes the morphological diversity injected by the random walk, yielding a similar drop ($-7.0$ seg AP). Diversity in crack length and branching is thus essential for generalisation. **A3** *Vanilla ControlNet.* Using the standard latent UNet without our topology branch reduces bbox AP by $2.6$ and seg AP by $3.3$, the largest single loss. Preserving connectivity during Mask→Image synthesis is therefore critical. **A4** *TC-ControlNet variants.* Feeding only TopoDownsample masks (no SPADE) or only SPADE Norm (no TopoDownsample) recovers part of the gain, but neither matches the full model.

In sum, every module, RIP, PSG, and the dual innovations of TC-ControlNet, contributes measurably; removing any of them degrades instance segmentation, while the complete CrackInstSynth pipeline **A5** achieves the best accuracy (**+13.2** seg AP over the unaugmented baseline **A0**).

### 4.2.5 RUNTIME–ACCURACY TRADE-OFF OF TOPODOWNSAMPLE

We study the effect of the MIP solver tolerance on accuracy and latency for TopoDownsample. Fig. 5 (one representative dataset) plots accuracy gain vs. max time limit under MIPGap $\in \{5, 2, 1, 0.5\}\%$. The curves show monotonic but saturating improvements: tightening the gap yields higher accuracy with diminishing returns. In practice, we select the tightest feasible gap under a given time budget; if the budget is exceeded, we *fallback* to bilinear downsampling to guarantee responsiveness.

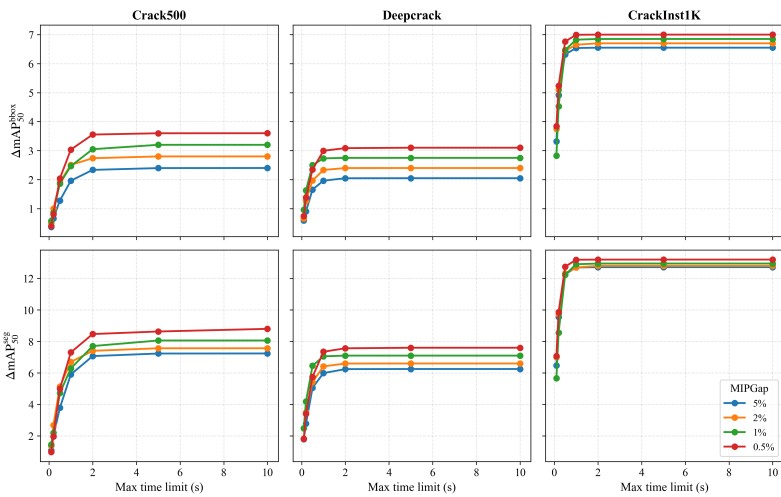

Figure 5: Runtime–accuracy trade-off of TopoDownsample. Accuracy gain vs. max time limit under different MIPGap settings on a representative dataset.

Across datasets and metrics, a moderate tolerance (MIPGap 1–2%) within a 1–2 s budget captures almost all of the attainable gains, while tighter settings incur disproportionate latency for marginal improvements.

## 5 CONCLUSION

We introduced **CrackInst1K** (high-resolution tunnel-crack data with per-instance masks) and **CrackInstSynth** (a topology-aware augmentation pipeline combining RIP, PSG, and TPGM/TC-ControlNet), which synthesizes realistic, topology-consistent image–mask pairs and consistently boosts instance segmentation performance. We will release both resources to support future work in structural health monitoring.

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

# A    APPENDIX

This appendix complements the main paper with six self-contained parts:

1. Section **Additional related work** A.1 show additional related work, including the recent works about topology-aware diffusion for curvilinear structures.

2. Section **Details of *CrackInst1K* Dataset** A.2 documents the new *CrackInst1K* dataset— its imaging pipeline, annotation protocol, statistics, example images, and anonymisation policy.

3. **Section More details about CrackInstSythn** A.3 give more Implementation details of CrackInstSythn.

4. Section **Rationale Behind *TC-ControlNet* Design** A.4 explains the design of *TC-ControlNet*, clarifying how TopoDownsample and SPADE jointly preserve crack topology and appearance.

5. Section **TopoDownsample: Formulation, Implementation & Theoretical Analysis** A.5 gives the full MIP formulation of TopoDownsample, solver details, qualitative comparisons, and a formal proof that the method preserves Betti numbers and the region-adjacency graph.

6. Section **Additional Details for Experiments** A.6 lists all hyper-parameters, hardware, prompts, detector settings, and evaluation metrics used in the experiments, followed by additional quantitative results.

7. Section **More visualizations** A.6.3 provides extra end-to-end visualisations, illustrating that the generated masks maintain perfect crack geometry across multiple rendering styles.

## A.1    ADDITIONAL RELATED WORK

### A.1.1    TOPOLOGY-AWARE DIFFUSION FOR CURVILINEAR STRUCTURES

Recent works inject explicit topological objectives into diffusion. TopoDiffusionNet Gupta et al. (2024) and TopoCellGen Xu et al. (2025) incorporate persistent-homology (PH) constraints to guide denoising toward target Betti profiles, improving topological faithfulness beyond appearance similarity. For linear networks, ControlTraj Zhu et al. (2024) enforces path-level constraints under diffusion to maintain global connectivity and branching structure. In medical imaging, a topology-aware conditional LDM preserves vascular connectivity and branching via PH-guided losses across views Demirci et al. (2025). Closer to cracks, semantic diffusion–based pavement synthesis improves realism and segmentation but does not explicitly regulate connectivity or non-self-intersection during generation Cano-Ortiz et al. (2024).

Unlike PH-loss–based approaches (e.g., TopoDiffusionNet) that condition on multi-object topology, we follow the semantics-to-image paradigm and focus on single-crack instance topology consistency. Concretely, we adopt a two-stage design that couples a physics-guided skeleton generator

| Original image | Annotation | Original image | Annotation |
| --- | --- | --- | --- |

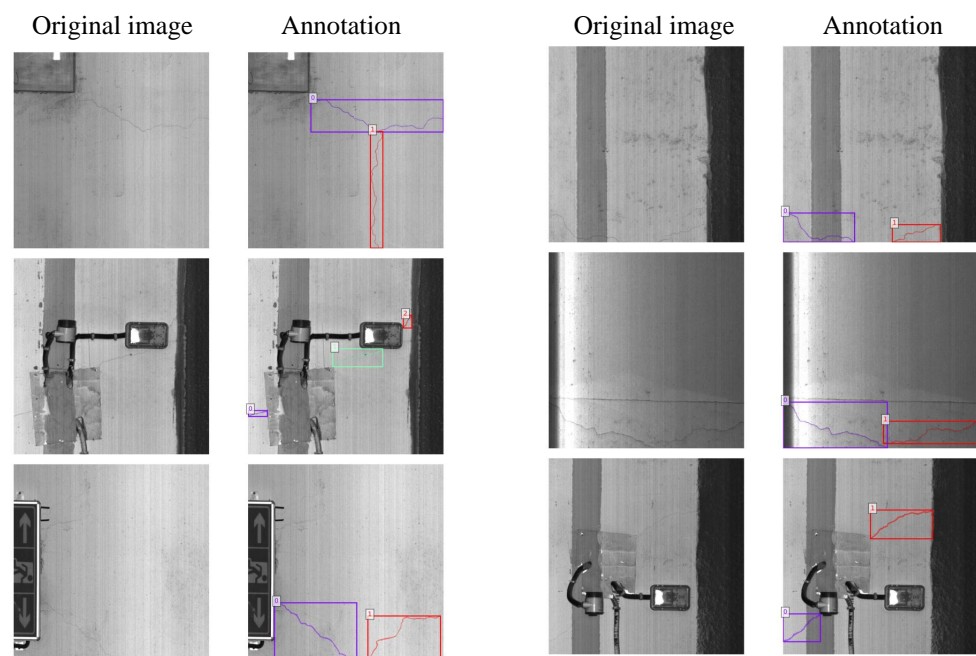

Figure 6: Representative samples from *CrackInst1K*. Each column pair shows the original image patch (left) and the corresponding instance annotation (right). Individual cracks are outlined with unique colours and numeric IDs, while background objects remain unlabelled to reflect real maintenance scenes.

(PSG) with a topology-preserving conditioning module, injecting crack-specific priors to yield more informative synthetic samples for downstream instance analysis.

## A.2 DETAILS OF *CrackInst1K* DATASET

### A.2.1 SCOPE AND MOTIVATION

*CrackInst1K* is a public benchmark for instance-level crack segmentation in civil-infrastructure imagery. It contains 1025 tunnel-lining patches (1024×1024 pixels), released in COCO format. The dataset supplies scale-aware, topology-preserving annotations for algorithms that must separate neighbouring cracks and trace fine branches.

### A.2.2 IMAGING AND PRE-PROCESSING PIPELINE

Images were captured in situ with a vehicle-mounted line-scan system. The rig maintains orthogonal viewing geometry and uniform illumination while travelling at approximately 3–5 km/h, producing raw stripes of 1000×7448 pixels that resolve cracks as thin as 0.29 mm. After acquisition, stripes were auto-stitched and frames containing visible cracks were retained. Each selected frame was cropped to square regions (800–3 000 pixels per side) and finally resized to 1024×1024 for release. Patches are split *per scene* to prevent leakage: 923 images for training and 102 for validation (90/10).

Fig. 6 gives a visual impression of the dataset, highlighting the thin, meandering geometry of cracks and the clutter commonly encountered inside tunnel environments.

### A.2.3 ANNOTATION PROTOCOL

Annotation followed a four-stage procedure:

1. **Polygon tracing:** crack contours were digitised as dense polygons (mean 140 vertices).

2. **Instance labelling:** every polygon receives a unique identifier; intersecting cracks are traced separately.

3. **Automatic sanity checks:** scripts flag self-intersections, duplicate vertices or masks that leak outside the canvas.

4. **Double-blind review:** two independent annotators correct flagged masks; a third reviewer resolves conflicts.

The final JSON stores each polygon, its bounding box, skeleton length and the physical pixel size.

### A.2.4 DATA ANONYMISATION AND AVAILABILITY

The raw imagery was collected over multiple years from several geographically distinct tunnels. During pre-processing, every exported $1024 \times 1024$ patch is a spatial mosaic drawn from different time stamps and camera poses. This strategy removes any location-specific patterns and prevents re-identification of the original infrastructure.

**Availability** – The dataset and scripts will be released upon acceptance to support reproducible research.

### A.3 MORE DETAILS ABOUT CRACKINSTSYTHN

### A.3.1 MORE DETILS ABOUT PSG

Algorithm 1 sketches the RandomWalk procedure.

---

**Algorithm 1** Physics-driven random walk within a rescaled bbox

---

1: **Input:** skeleton $\mathbf{S}$, growth window $\hat{\mathcal{B}}$, parameters $(k, m, max, \ell, \theta)$
2: Initialise queue $Q$ with $k$ random start pixels on $\mathbf{S}$
3: $\mathbf{S}^{\text{new}} \leftarrow \mathbf{S}$
4: **while** $Q \neq \varnothing$ **do**
5:     Pop current point $(x, y, d)$ where $d$ stores the incoming direction
6:     **if** length$(\mathbf{S}^{\text{new}})/\operatorname{diag}(\hat{\mathcal{B}}) \geq m$ **or** steps $> max$ **then**
7:         **continue**
8:     **end if**
9:     Sample turning angle $\Delta\phi \sim \mathcal{U}(-\theta, \theta)$
10:     $d' \leftarrow d + \Delta\phi$;  $(x', y') \leftarrow (x, y) + \ell(\cos d', \sin d')$
11:     **if** $(x', y') \in \hat{\mathcal{B}}$ **and** not occupied **then**
12:         Add $(x', y')$ to $\mathbf{S}^{\text{new}}$ and push $(x', y', d')$ to $Q$
13:     **end if**
14: **end while**
15: **return** $\mathbf{S}^{\text{new}}$

---

Compared with pure geometric jittering, PSG implements a physics-based simulation of crack growth by carefully tuning the parameters of a bounded random walk, thereby increasing instance-level information content. In practice, PSG adds 35% new skeleton pixels per instance in average, providing diverse yet physically credible conditional masks for the subsequent Topology-Preserving Generation Module (TPGM).

### A.3.2 MORE DETAILS ABOUT TPGM

TPGM $\mathcal{F}: \mathbf{S}_{\text{PSG}} \longmapsto (\mathbf{M}_{\text{WA}}, \mathbf{I})$ as much as possible meet the following conditions:

- *Intra-instance topology:* for every instance $c > 0$, the region $\{p : \mathbf{M}_{\text{WA}}(p) = c\}$ forms a *single* 8-connected component (i.e., $\beta_0 = 1$, no holes).

- *Inter-instance separation:* $\forall p$ and $\forall c \neq d$, it never holds that $\mathbf{M}_{\text{WA}}(p) = c \wedge \mathbf{M}_{\text{WA}}(p) = d$ (regions are mutually exclusive).

- *Pixel-wise consistency:* cracks in $\mathbf{I}$ correspond one-to-one with labels in $\mathbf{M}_{\mathrm{WA}}$; in particular, the crack support in $\mathbf{I}$ exactly matches $\{p : \mathbf{M}_{\mathrm{WA}}(p) \neq 0\}$, and for each $c > 0$ the rendered crack $c$ coincides with $\{p : \mathbf{M}_{\mathrm{WA}}(p) = c\}$.

### A.3.3 More details about Stage 1: Skeleton→Mask Diffusion

*Training pairs.* We construct training pairs $(\mathbf{S}_{\mathrm{sk}}, \mathbf{M}_{\mathrm{gt}})$ from crack segmentation datasets by skeletonizing each ground-truth mask $\mathbf{M}_{\mathrm{gt}}$ with Zhang–Suen thinning Lam et al. (1992) to obtain $\mathbf{S}_{\mathrm{sk}}$. At inference time, $\mathbf{S}_{\mathrm{PSG}}$ (from PSG) replaces $\mathbf{S}_{\mathrm{sk}}$ as the conditioning input.

*Hyperparameters.* We follow the original SDM setup Wang et al. (2022); Lei et al. (2024a): cosine $\beta_{1:T}$ schedule, $T{=}1000$ training steps, DDIM 20 sampling steps, UNet depth 4 with 128 base channels, AdamW (learning rate $1{\times}10^{-4}$, weight decay $1{\times}10^{-2}$). We train for 150k iterations with batch size 16.

*Sampling.* Given a new skeleton mask, we sample $\mathbf{x}_T \sim \mathcal{N}(\mathbf{0}, \mathbf{I})$ and run the DDIM solver for 20 steps to obtain $\hat{\mathbf{M}}_{\mathrm{WA}}$. A channelwise argmax yields the discrete width-aware map that feeds Stage 2.

### A.4 Rationale Behind *TC-ControlNet* Design

**Background.** ControlNet Zhang et al. (2023) augments Stable Diffusion Rombach et al. (2022) with an extra condition $c$ (for example, a segmentation map) to guide generation. The input mask $c \in \mathbb{R}^{H \times W \times 3}$ is first embedded by a small CNN, $h = E_c(c) \in \mathbb{R}^{\frac{H}{8} \times \frac{W}{8} \times 4}$, and added to the noisy latent $z_t$ before entering the trainable branch of a U-Net copied from the frozen backbone.

**Problem.** GroupNorm layers inside the U-Net average spatial statistics and tend to erase fine semantic cues Park et al. (2019); Lei et al. (2024b). Moreover, naively downsampling thin-object masks to the latent resolution (1/8, 1/16, 1/32) breaks connectivity and alters topology, as shown in the main paper.

**Solution.** **TC**-**ControlNet** addresses both issues with two design choices:

1. **TopoDownsample**. The binary crack mask is reduced to three latent scales by solving a small mixed-integer program that preserves Euler characteristic and region-adjacency, avoiding the aliasing artefacts of convolutional or interpolation-based resizing.

2. **SPADE feature modulation.** Each encoder block replaces GroupNorm with SPADE Park et al. (2019), using the topology-safe masks from TopoDownsample as spatially varying scale and bias to reinject crack geometry lost during normalisation.

**Illustration.** Fig. 7 demonstrates the effect of the two modules on two representative crack masks. Column (a) shows the original $512 \times 512$ mask; columns (b)–(d) display the TopoDownsample outputs at resolutions 64, 32, and 16. Even at 1/32 resolution, the skeleton remains connected and free of spurious bridges. Column (e) gives the final photorealistic image generated by TC-ControlNet; the crack paths precisely match the conditioning masks, confirming that topology is retained through both the latent UNet and the RGB decoder.

**Fallback strategy.** If the mixed-integer solver fails to find a feasible assignment at a given scale (rare in practice), the algorithm falls back to bilinear interpolation for that scale only, ensuring continuity of the generation pipeline.

By combining topology-aware downsampling with spatially adaptive modulation, TC-ControlNet mitigates semantic dilution and delivers superior topological fidelity compared with vanilla Control-Net, as validated quantitatively and qualitatively in the main paper.

### A.5 TopoDownsample: Formulation, Implementation & Theoretical Analysis

Simply resizing a thin–object mask from $512{\times}512$ to latent grids (64, 32, 16) with nearest, bicubic or pooling interpolation breaks connectivity and may introduce spurious holes. *TopoDownsample*

(a) (b) (c) (d) (e)

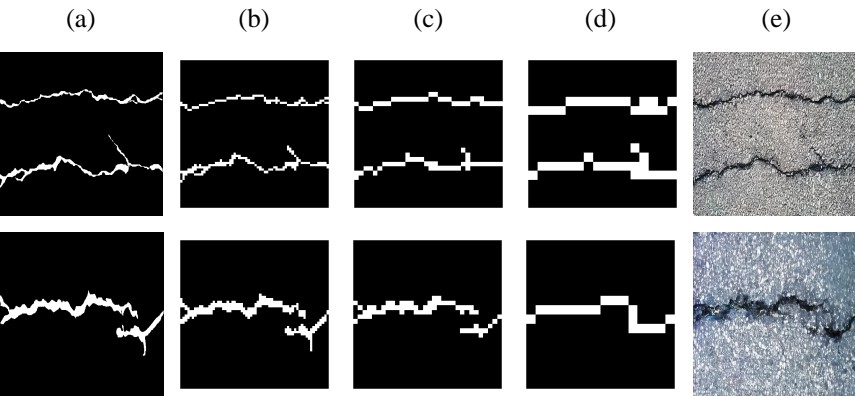

Figure 7: Effect of TopoDownsample and SPADE in TC-ControlNet. (a) Original $512 \times 512$ crack mask; (b)–(d) topology-preserving masks at resolutions 64, 32, and 16; (e) asphalt-style image synthesised by TC-ControlNet.

addresses this by formulating down-sampling as a compact mixed-integer programme (MIP) whose feasible set contains *only* pixel assignments that preserve foreground–background topology Chen & Peng (2024).

### 1. PROBLEM SET-UP

Let $c \in \{0,1\}^{H \times W}$ be the input binary mask and $c^{(k)}$ its coarse version at scale $k \in \{0,1,2\}$, height $H_k = H/2^{3-k}$. Each pixel of $c^{(k)}$ aggregates an $s_k \times s_k$ block of $c$, where $s_k = H/H_k$.

**Decision variables.**

- $x_{m,i,j}^{(k)} \in \{0,1\}$: macro-pixel $(i,j)$ belongs to component $m$.
- $z_v^{(k)} \in \{0,1\}$: vertex $v$ activates a valid corner configuration from the catalogue in Chen & Peng (2024).
- $l_v^{(k)} \in \{0,1\}$: unique terminal flag closing each boundary loop.

### 2. OBJECTIVE

$$\max_{x,z,l} \sum_{m,i,j} w_m \, S_m(i,j) \, x_{m,i,j}^{(k)},$$

where $S_m(i,j)$ is the component–block overlap score and $w_m = 2$ for foreground, 1 for background Chen & Peng (2024).

### 3. CONSTRAINTS

1. **Exclusivity:** $\sum_m x_{m,i,j}^{(k)} = 1 \quad \forall (i,j)$.

2. **Component survival:** $\sum_{(i,j) \in \mathcal{R}_m} x_{m,i,j}^{(k)} \geq 1 \quad \forall m$.

3. **Anti-diagonal (background):** $x_{m,i,j}^{(k)} + x_{m,i+1,j+1}^{(k)} \leq 1$ for every background component $m$.

4. **Boundary continuity:** $z_v^{(k)} \Rightarrow \bigvee_{v' \in \mathcal{N}(v)} z_{v'}^{(k)}$.

5. **Loop closure:** $\sum_{v \in \mathcal{V}_b} l_v^{(k)} = 1$ for each boundary $b$.

Items 1–3 ensure a valid label map; 4–5 force every interface to form a single closed 8-connected curve, thereby preserving Euler characteristic $\chi = \beta_0 - \beta_1$ Kong & Rosenfeld (1989).

## 4. COMPONENT AND CORNER ENUMERATION

Foreground components are extracted with 8-connectivity, background with 4-connectivity—the standard "complementary" pairing that avoids paradoxes in digital topology Rosenfeld (1979). At each grid vertex we test the twelve corner templates of Chen & Peng (2024); invalid patterns are discarded, shrinking the MIP.

## 5. SOLVER DETAILS

The MIP is implemented in C++ and solved with Gurobi Gurobi Optimization, LLC (2024), while a Python port is provided for visualisation. A greedy warm start assigns each macro-pixel to the component covering the largest area. If a scale is infeasible (rare; $< 0.3\%$ at $16 \times 16$), TopoDownsample falls back to bilinear interpolation for that scale only.

## 6. QUALITATIVE COMPARISON

Fig. 8 contrasts TopoDownsample with five baselines (nearest, bicubic, pooling, ACN, dilation). Only our method preserves the crack's topology at $64$, $32$ and $16$ pixels.

## 7. THEORETICAL ANALYSIS

We now prove that any feasible MIP solution preserves the Region Adjacency Graph (RAG) Stockman & Shapiro (2001) and the Betti numbers $\beta_0$ (components) and $\beta_1$ (holes).

**Lemma 1.** (*Component preservation*) *The down-sampled mask has exactly the same number of $8$-connected foreground and $4$-connected background components as the original; hence $\beta_0$ is unchanged.*

*Proof.* Component survival (Constraint 2) forbids disappearance. Anti-diagonal plus exclusivity (Constraints 1–3) forbid two distinct components from touching, preventing mergers. If an original component attempted to split, its boundary would fragment into two closed curves, violating the single-loop requirement (Constraint 5). Thus one-to-one correspondence of components holds. □

**Lemma 2.** (*Hole preservation*) *Every original hole persists in the down-sampled mask and no new hole is created; therefore $\beta_1$ is unchanged.*

*Proof.* A hole is a $4$-connected background component fully enclosed by a foreground boundary. Lemma 1 guarantees the hole itself survives. Boundary continuity (Constraint 4) and unique-loop (Constraint 5) keep its enclosing Jordan curve intact, preventing the hole from leaking into exterior background. Because the MIP introduces no additional black–white adjacencies, no extra closed curve—and hence no extra hole—can arise. □

**Lemma 3.** (*RAG preservation*) *The Region Adjacency Graph of the down-sampled mask is isomorphic to that of the input.*

*Proof.* By Lemma 1, nodes (regions) correspond one-to-one. For each original black–white pair, Constraint 5 instantiates exactly one closed boundary loop, producing the same edge in the output RAG. Pairs not adjacent originally remain separated by at least one pixel, and no new edge appears because no new boundary loop is allowed. □

**Theorem 1.** (*Topology preservation*) *Any feasible solution of the TopoDownsample MIP preserves $\beta_0$, $\beta_1$, and the entire RAG of the binary mask.*

*Proof.* Immediate from Lemmas 1, 2 and 3. □

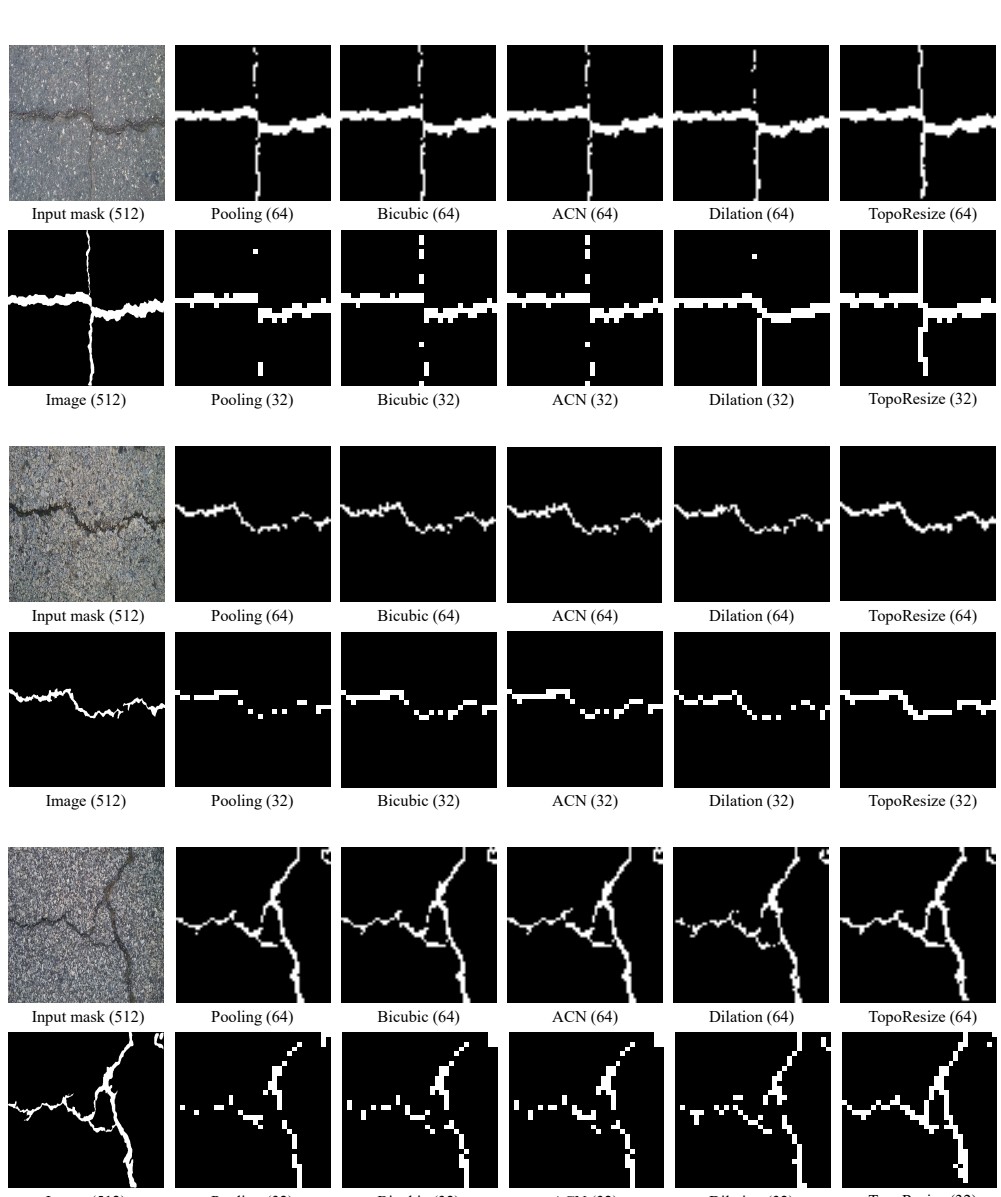

Figure 8: TopoDownsample versus conventional down-sampling methods on two crack masks.

## A.6 ADDITIONAL DETAILS FOR EXPERIMENTS

### A.6.1 EXPERIMENTAL SETUP

**Hyper-parameter settings**

All experiments were run on a workstation equipped with two NVIDIA RTX A6000 GPUs (48 GB each), an AMD EPYC 7513 CPU, and 256 GB RAM. Gurobi 11.0 is used for all MIP optimisations.

Unless stated otherwise, all hyper-parameters below are shared across *CrackInst1K*, *DeepCrack* and *CRACK500*.

- **Canvas tiling (RIP).** The $1024{\times}1024$ canvas is partitioned into four $512{\times}512$ quadrants; $n \sim \mathcal{U}\{1, 2, 3\}$ instances are sampled without replacement and pasted into randomly permuted quadrants. Source boxes are translated uniformly inside each quadrant, avoiding overlap.
- **BBox expansion for PSG.** For every seed instance the bounding box is isotropically enlarged by an independent scale factor $\alpha_x, \alpha_y \sim \mathcal{U}(1, 1+s)$ with $s = 0.6$. The enlarged window defines the admissible region for random walks.
- **Random-walk skeleton growth (PSG).** Number of starting points $k \sim \mathcal{U}\{0, \dots, 4\}$; maximum crack-pixel ratio $m = 0.8$; step length 2 px; turning angle $\pm 30°$; step budget per start point `min_steps = 30`, `max_steps = 100`.
- **File resolution.** All intermediate masks are kept at $1024{\times}1024$; diffusion stages operate at $512{\times}512$ and outputs are up-sampled back to 1024 if needed using Lanczos.

**Synthetic-data generation**

- **SDM (Skeleton→Mask).** UNet depth 4, base channels 128; cosine $\beta$ schedule, $T{=}1\,000$ training steps, DDIM 20 sampling steps; learning rate $1{\times}10^{-4}$, AdamW with weight decay $1{\times}10^{-2}$, batch size 16 for 150 k iterations.
- **TC-ControlNet (Mask→Image).** Frozen SD 1.5 backbone, ControlNet channel multiplier 0.5, TopoDownsample at scales $64/32/16$, SPADE injection at every encoder block, classifier-free guidance scale 7.5, DDIM 20 inference steps.
- **Prompt settings.** For the Crack500 dataset, the prompt settings from COSTG Lei et al. (2024b) were used. For the Deepcrack and CrackInst1K datasets, the prompt templates are as follows: *An image of cracks in a tunnel lining (road pavement); CrackInst1K dataset (Deepcrack dataset); there are(is) k cracks(s) in this image.* Here $k = \{1, 2, 3\}$ is the number of crack instances randomly placed during RIP.
- **Augmentation budget.** Each training split is expanded to exactly $5\times$ its original size (Table 2, main paper). Synthetic images are saved at $1024{\times}1024$, then centre-cropped to $1024{\times}1024$ before training.

**Downstream instance segmentation**

The goal of this experiment group is to measure how much **CrackInstSynth** augments improve *downstream* crack instance segmentation. Unless noted otherwise, every detector is trained *twice*: (i) on the original real-image training split and (ii) on the augmented set (real + synthetic at $5\times$ scale). All hyper-parameters below are kept identical between the two runs so that any accuracy difference can be attributed solely to the synthetic data.

- **Mask R-CNN He et al. (2017).** ResNet-50-FPN backbone initialised from COCO; SGD (lr 0.02, momentum 0.9, wd $1{\times}10^{-4}$); linear warm-up 1k iters, step drops at epochs 60 and 80; 100 epochs, global batch size 8 (CrackInst1K) or 12 (DeepCrack / CRACK500); data aug. = random flip (p 0.5) + scale jitter $[0.8, 1.2]$; evaluation with COCO AP at IoU 0.50 using `coco_eval.py` in Detectron2.
- **Cascade Mask R-CNN Cai & Vasconcelos (2018).** R50-FPN, three-stage cascade; other settings identical to Mask R-CNN; configuration follows the official Detectron2 recipe.
- **Mask2Former Cheng et al. (2022).** Swin-L backbone, $2\times$ LR schedule (100 epochs on our datasets), AdamW optimiser with parameters from the original paper; all other hyper-parameters unchanged.

- **CondInst Tian et al. (2020).** R50-FPN; trained with default 3× schedule in Detectron2 but capped at 100 epochs for parity.

- **SOLOv2 Wang et al. (2020).** ResNet-101-FPN; we adopt the authors' public configuration from MMDetection 3.3; total epochs 100.

- **QueryInst Fang et al. (2021).** Swin-T backbone; default learning-rate schedule from the paper; batch size 8 due to GPU memory.

Loss weights, anchor settings, and post-processing remain exactly as in the respective reference implementations; no detector-specific tuning is performed.

**Image-quality evaluation**

- **FID.** Computed on 10k CrackInstSynth images *vs.* the entire real training split of the same dataset; Inception-V3 *pool3* features, TORCH-FID.

- **mIoU and Betti errors.** Each *instance* mask is collapsed to a *binary* crack-vs-background map before evaluation so that topology metrics reflect true foreground connectivity, independent of instance IDs. A robust U-Net Lei et al. (2024b), predicts binary masks for 2k synthetic images; results are compared to ground truth to obtain mIoU as well as absolute Betti-number errors ($|\Delta\beta_0|$, $|\Delta\beta_1|$). Connected-component analysis uses SCIKIT-IMAGE measure.label with 8-connectivity for foreground and 4-connectivity for background, matching the TopoDownsample convention.

A.6.2 MORE EXPERIMENTAL RESULTS

**More detectors performance on CrackInstSynth**

Table 4 reports the $\text{mAP}_{50}^{\text{seg}}$ achieved by six detectors on three benchmarks, *with* and *without* Crack-InstSynth augmentation. All models gain accuracy, with TC-ControlNet data giving the largest boost on the most data-hungry detector (Mask R-CNN).

Table 4: Segmentation mAP50 before and after adding CrackInstSynth training data.

| Detector | CrackInst1K | | DeepCrack | | CRACK500 | |
|---|---|---|---|---|---|---|
| | Base | +Synth | Base | +Synth | Base | +Synth |
| Mask R-CNN | 70.1 | 83.3 | 84.6 | 92.2 | 75.4 | 84.2 |
| Cascade Mask R-CNN | 71.5 | 82.0 | 85.3 | 91.0 | 76.0 | 83.6 |
| Mask2Former | 78.0 | 86.1 | 89.4 | 94.0 | 80.2 | 87.1 |
| CondInst | 69.4 | 78.4 | 83.2 | 88.5 | 73.0 | 80.6 |
| SOLOv2 | 65.0 | 73.2 | 79.8 | 85.1 | 68.7 | 75.8 |
| QueryInst | 79.3 | 86.3 | 78.7 | 93.7 | 80.1 | 85.0 |

**Cross-dataset generalisation**

To evaluate domain robustness, we adopt a leave-one-dataset-out protocol: the detector is trained on a single *source* set (with or without CrackInstSynth) and tested *zero-shot* on the remaining two *target* sets. Table 5 shows that synthetic training data consistently improve segmentation accuracy by **6–9** across all six source–target pairs.

Synthetic samples derived from the *source* domain thus transfer positively, even without any access to *target* images, confirming CrackInstSynth's value for real-world deployment.

**Sensitivity to the synthetic-to-real ratio**

Fig. 9 shows how Mask R-CNN $\text{mAP}_{50}^{\text{seg}}$ evolves when the synthetic-to-real ratio increases from 1:1 to 50:1. The trend is consistent across all three benchmarks: accuracy climbs rapidly up to a 5:1 ratio and saturates thereafter. Consequently, we fix 5× synthetic augmentation for all main-paper experiments.

Table 5: Cross-dataset generalisation of Mask R-CNN between datasets. $\text{mAP}_{50}^{\text{seg}}$.

| Source | Target | Base | +Synth |
|--------|--------|------|--------|
| CrackInst1K | DeepCrack | 60.2 | **67.5** |
| | CRACK500 | 58.9 | **65.1** |
| DeepCrack | CrackInst1K | 62.1 | **69.3** |
| | CRACK500 | 61.0 | **68.2** |
| CRACK500 | CrackInst1K | 58.0 | **64.8** |
| | DeepCrack | 59.1 | **66.4** |

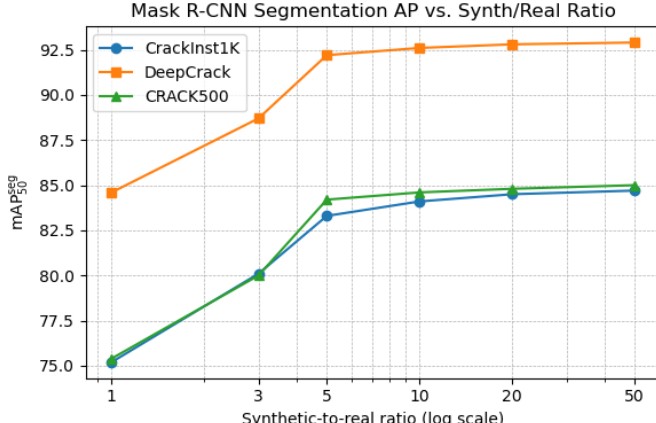

Figure 9: Mask R-CNN segmentation AP versus the amount of synthetic data, plotted on a log-scaled $x$-axis.

### A.6.3 MORE VISUALIZATIONS

Fig. 10 shows skeleton→mask→image examples with four material domains. Columns (a)–(b) illustrate that physics-driven growth and the Skeleton2Mask diffusion recover realistic widths and branching. Columns (c)–(f) demonstrate prompt-driven style transfer: *tunnel*, *pavement*, *brick*, and *concrete*. Across rows, crack pixels remain aligned to the same mask and connectivity is unchanged, indicating *label-faithful, multi-style* synthesis that supports our cross-dataset generalization experiments.

### A.6.4 PSG QUANTITATIVE ALIGNMENT: FULL STATISTICS

To demonstrate that PSG-generated masks match real masks in key *curvilinear* and *topological* statistics beyond visuals, we report full distribution distances (Wasserstein $W_1$ and Kolmogorov–Smirnov) and robust-location summaries (medians/IQRs).

| Feature | $W_1$ | KS stat | KS $p$-value | $n_{\text{real}}$ | $n_{\text{psg}}$ |
|---------|-------|---------|--------------|-------------------|------------------|
| length_mean | 13.3595 | 0.0300 | 0.688 | 1134 | 1134 |
| tortuosity_mean | 0.0143 | 0.4109 | $4.58\times10^{-86}$ | 1134 | 1134 |
| turning_std | 0.00512 | 0.0591 | 0.0382 | 1134 | 1134 |
| curvature_std | 0.00830 | 0.3298 | $5.51\times10^{-55}$ | 1134 | 1134 |
| deg3p_share | 0.00633 | 0.0926 | $1.19\times10^{-4}$ | 1134 | 1134 |
| $\beta_0$ | 0.0203 | 0.00706 | 1.000 | 1134 | 1134 |
| $\beta_1$ | 0.0176 | 0.0106 | 1.000 | 1134 | 1134 |

Table 6: REAL vs. PSG: Wasserstein $W_1$ and KS statistics across curvilinear/topology features.

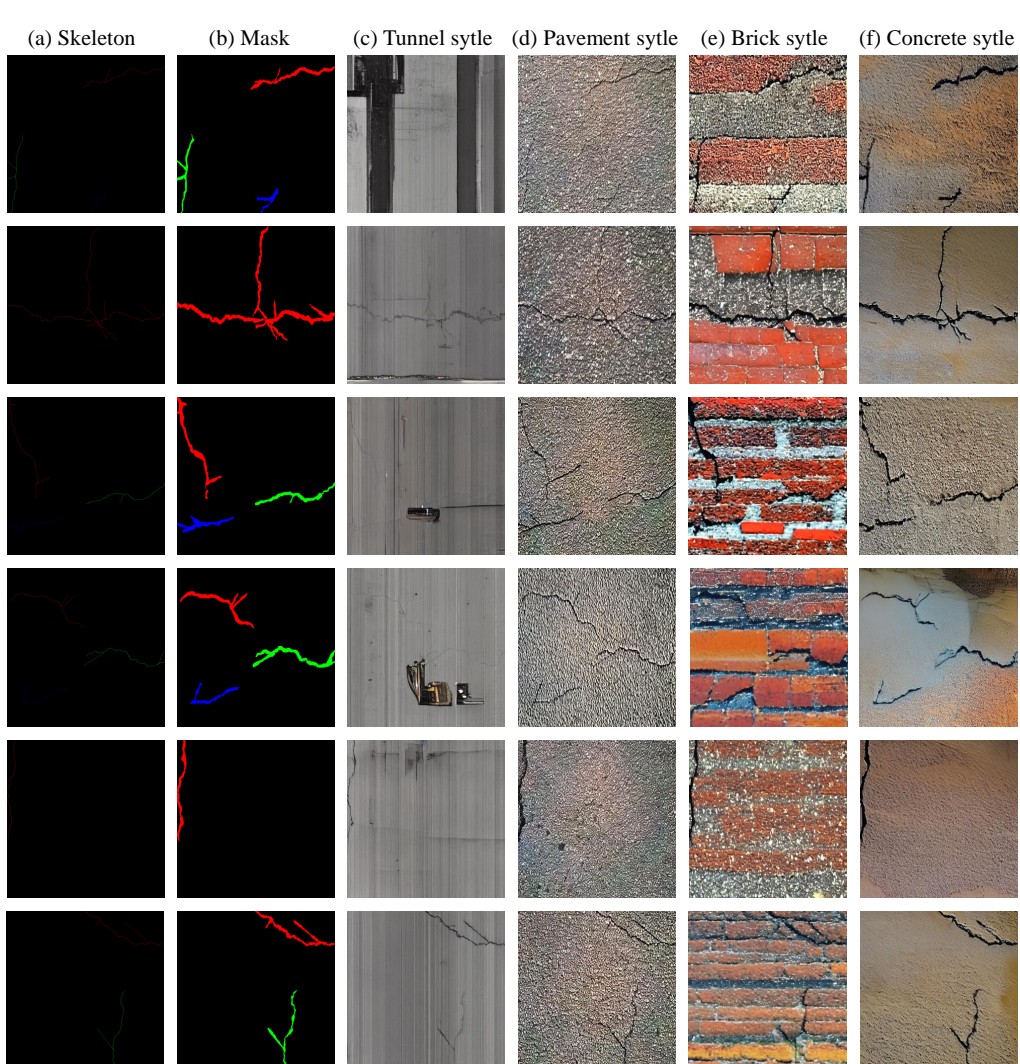

(a) Skeleton (b) Mask (c) Tunnel sytle (d) Pavement sytle (e) Brick sytle (f) Concrete sytle

Figure 10: **Additional visualizations across materials (CrackInstSynth).** Columns: (a) *Skeleton* after physics-driven growth; (b) *Mask* produced by the Skeleton→Mask diffusion; (c) *Tunnel* style; (d) *Pavement* style; (e) *Brick* style; (f) *Concrete* style. The same mask (instance IDs in colour) is rendered into different materials by switching text prompts. TC-ControlNet transfers background appearance while preserving per-instance topology and width.

In Tab. 6, $W_1$ distances are small across all features, and KS statistics are low for most metrics; $\beta_{0,1}$ match particularly well (KS $p$=1.0). For tortuosity and curvature, KS $p$-values are tiny due to the large sample size, but the corresponding $W_1$ magnitudes remain very small, indicating negligible practical shift.

| Feature | REAL | | | PSG | | |
|---|---|---|---|---|---|---|
| | median | p25 | p75 | median | p25 | p75 |
| length_mean | 208.79 | 74.48 | 558.36 | 207.51 | 80.04 | 557.89 |
| tortuosity_mean | 1.000 | 1.000 | 1.000 | 1.000 | 1.000 | 1.027 |
| turning_std | 0.5391 | 0.5173 | 0.5567 | 0.5364 | 0.5140 | 0.5600 |
| curvature_std | 0.3441 | 0.3409 | 0.3453 | 0.3425 | 0.3328 | 0.3530 |

Table 7: REAL vs. PSG: medians and IQRs (p25–p75). Small absolute $W_1$ and aligned robust summaries indicate close distributional agreement.

In Tab. 7, medians and IQRs of key curvilinear statistics are closely aligned between REAL and PSG; deviations are minor and directionally consistent with the small $W_1$ in Tab. 6.

Tables 6 and 7 jointly show that PSG preserves the distribution of curvilinear (length, tortuosity, turning/curvature) and topological ($\beta_{0,1}$) descriptors: effect sizes ($W_1$) are small and robust summaries (medians/IQRs) align closely with REAL data. While some KS tests become significant under large $n$, the practical shifts are negligible. These results substantiate the claim that PSG is *physics-consistent* beyond visual plausibility and does not distort the data-generating geometry.

### A.6.5 PSG SENSITIVITY AND RESOLUTION-AWARE SCALING

To clarify what matters to tune in PSG and how to port settings across resolutions, we sweep key parameters at $1024^2$ and report a simple resolution rule with normalized geometry metrics.[1]

**Parameter sweep** (@$1024^2$, $m$=0.10). Defaults: $k$=4, $\ell$=2 px, max=200 iters (max=$100 \cdot \ell$), $\theta \in \{\pm 15°, \pm 30°, \pm 45°\}$.

| Param (values) | Best | Aggregate $W$ range | $\Delta$ vs. default | # within +5% | Note |
|---|---|---|---|---|---|
| $k$ (2,4,8) | 4 | $13.3904 \rightarrow 13.7421$ | $+0.0\% \rightarrow +2.6\%$ | 3/3 | Highly insensitive |
| max (100,200,300) | 200 | $13.3904 \rightarrow 13.4400$ | $+0.0\% \rightarrow +0.4\%$ | 3/3 | Runtime knob; stable |
| $\ell$ px (1,2,3) | 2 | $13.3904 \rightarrow 14.8500$ | $+0.0\% \rightarrow +10.9\%$ | 2/3 | Affects lengths |
| $\theta$ ($\pm 15°, \pm 30°, \pm 45°$) | $\pm 30°$ | $13.3904 \rightarrow 14.5800$ | $+0.0\% \rightarrow +8.9\%$ | 2/3 | Affects curvature |

Table 8: PSG sensitivity at $1024^2$: most parameters are robust; $\ell$ and $\theta$ modulate geometry as expected.

In Tab. 8, varying $k$ or max has negligible effect ($\leq 2.6\%$ and $\leq 0.4\%$ on the aggregate $W$), whereas $\ell$ and $\theta$ control geometry (length/curvature) with still moderate shifts; most settings remain within +5% of the default.

**Resolution scaling rule.** Let $s = \frac{\text{target}}{1024}$; keep $(k, m)$ fixed; scale $\ell = 2s$ px and max $= 100 \cdot \ell$.

| Resolution (rule-scaled) | Normalized Aggregate $W$ | $\Delta$ vs. $1024^2$ | Note |
|---|---|---|---|
| $512^2$ ($s$=0.5, $\ell$=1, max=100) | 0.0449 | $+1.9\%$ | Normalized by size |
| $1024^2$ ($s$=1.0, $\ell$=2, max=200) | 0.04399 | $0.0\%$ | Baseline |
| $2048^2$ ($s$=2.0, $\ell$=4, max=400) | 0.0434 | $-1.3\%$ | Normalized by size |

Table 9: Resolution-aware scaling keeps geometry statistics stable after size normalization.

In Tab. 9), after size normalization, the aggregate geometry drift remains within $\pm 2\%$ from $512^2$ to $2048^2$, indicating that the linear rule preserves the distributional shape across resolutions.

---

[1]Aggregate metric: sum of 1D Wasserstein distances over length_mean, tortuosity_mean, turning_std, curvature_std, deg3p_share; lower is better.

For practical tuning, prioritize $\ell$ and $\theta$; the defaults $k=4$ and max$=200$ are sufficient. The simple scaling rule ($\ell=2s$, max$=100\ell$) maintains stable geometry statistics across resolutions while keeping the parameterization minimal.

