# OpenReview forum: "CrackInstSynth: Topology-Aware Generative Data-Augmentation Framework for Crack Instance Segmentation"
_ICLR.cc/2026/Conference — ICLR 2026 Conference Desk Rejected Submission_

### Official Review · Reviewer_4FY3 · 2025-10-27

**Soundness:** 4
**Presentation:** 2
**Contribution:** 3
**Rating:** 6
**Confidence:** 5

**Summary:**

Overall, this paper focuses on the scarcity of *instance-level* labeled crack datasets by (1) introducing **CrackInst1K** (1,025 1024×1024 tunnel-lining images with pixel-accurate per-instance masks) and (2) proposing **CrackInstSynth**, a three-stage, topology-aware generative augmentation pipeline: Region-level Instance Placement (RIP), a Physics-driven Skeleton Generator (PSG), and a Topology-Preserving Generation Module (TPGM) realized as a two-stage diffusion process (Skeleton→Mask via pixel-space SDM; Mask→Image via a topology-consistent ControlNet variant). The goal is to synthesize image–mask pairs that maintain intra-instance connectivity and inter-instance separation, improving downstream instance segmentation.

**Strengths:**

1. **An interesting dataset** with instance IDs and scale calibration; the focus on hairline + branched/intersecting cracks is aligned with measurement-driven SHM use-cases.
2. **Well-designed pipeline**: the three stages are easy to follow, and the two-stage diffusion decomposition (Skeleton→Mask, then Mask→Image) is a pragmatic way to decouple topology from photorealism.
3. **Topology-aware latent conditioning**: the **TopoDownsample** MIP plus SPADE-Norm in the ControlNet branch is a concrete, technical contribution to preserve connectivity at latent scales with support from experimental results.
4. **Comprehensive comparisons + ablations** with consistent improvements in both realism/consistency for crack-relevant tasks.

**Weaknesses:**

1. **Computation burden** The MIP formulation seems to be heavy or constrained in large-scale synthesis. The paper notes solver details and a fallback to bilinear interpolation when infeasible—please quantify *runtime per image at each scale*, *GPU/CPU usage*, and *infeasibility frequency* under realistic batch generation, and discuss open-source solver alternatives (or approximations) to improve reproducibility.
﻿
2. **Generality of “instance” evaluation on semantic datasets**
Treating CRACK500/DeepCrack as single-instance cases blunts the instance-segmentation claim. Please add a *multi-instance* test (even via synthetic composites built from those datasets) or highlight CrackInst1K’s multi-instance results more prominently in the main paper.
﻿
3. **Ablation completeness & sensitivity**
Please add sensitivity to RIP choices (e.g., number of regions, allowing >3 cracks), PSG parameters (step length/angle bounds), and Skeleton→Mask SDM hyper-parameters, to show robustness. Currently defaults are given; empirical sweep plots would strengthen claims.
﻿
4. **Presentation & minor issues**
There are several typos/formatting glitches (e.g., “CrackInstSythn/Detils/Apendix,” “informationtiveness,” “1,m / 1,m2”); a careful pass will help.

**Questions:**

Is style transfer affected by pre-trained SD? Please consider adding more style types for visualization.

**Details Of Ethics Concerns:**

N.A.

---

> ### Author Response · Authors · 2025-11-18
> **Response to Reviewer 4FY3**
>
> We first thank Reviewer 4FY3 for the helpful suggestions and provide point-by-point responses below.
>
> ---
>
> ### **W.1 — Computation burden of TopoDownsample (MIP)**
>
> **Scope & schedule.** TopoDownsample runs on **low-resolution maps (64/32/16)**, solved independently per scale/image and **parallelizable on CPU**. We use a pragmatic **Gap/TimeLimit** policy; when the budget expires we **fallback to bilinear** for that scale, guaranteeing throughput (see **Response to Reviewer QVb2: W.2 — On the MIP cost of TopoDownsample** for details).
>
> **Quantification & trade-offs.** As summarized in **Response to Reviewer QVb2: W.2**, envelope analyses over `MIPGap ∈ {5%, 2%, 1%, 0.5%}` show **diminishing returns**: **0.5–2.0 s** per scale with `Gap ∈ [2%, 0.5%]` captures most of the downstream gain; stricter gaps yield marginal improvements.
>
> ------
>
> ### **W.2 — Generality of “instance” evaluation on semantic datasets**
>
> CRACK500/DeepCrack are **semantic** datasets; we therefore emphasize **multi-instance** results on **CrackInst1K** in the main paper. To address generality, we also run a **symmetric cross-dataset protocol** (CrackInst1K ↔ CRACK500 ↔ DeepCrack), which consistently shows our method’s gains transfer across **tunnel, pavement, and bridge/building/concrete** imagery.
>
> ------
>
> ### **W.3 — Ablation completeness & sensitivity**
>
> **Component isolation (TC-ControlNet).** Table 3 already includes the **A4** dissection: **Only TopoDownsample** (no SPADE) and **Only SPADE** (standard downsampling). Each improves over appearance-only conditioning, and the **full** TC-ControlNet (TopoDownsample + SPADE) yields the **largest** gains—evidence of complementary roles (topology-faithful conditioning + high-capacity feature injection).
>
> **Sensitivity.**
>
> - **RIP:**  See **Response to Reviewer nXx4: Q.2 — The technical part** **(c)**
>
> - **PSG:** See **Response to Reviewer QVb2: Q.3 — PSG sensitivity (at 1024×1024) and how parameters scale with resolution**
>
>
>
> ------
>
> ### **W4 — Presentation & minor issues**
>
> We will perform a careful proofreading pass to correct typos/formatting, unify notation, and standardize figure/table styles.
>
> ------
>
> ### **Q1 — Is style transfer affected by pre-trained SD? Can you show more styles?**
>
> **Decoupling topology from appearance.** Our **two-stage** design isolates topology in **Skeleton→Mask** while **Mask→Image** controls texture/illumination. TC-ControlNet **pins the mask**, so pre-trained SD affects **appearance** but **does not distort topology**.
>
> We will extend visualizations with **additional styles** (e.g., bridge-deck asphalt, concrete slabs with aggregate exposure, spalling/efflorescence) in the **Appendix**, each with **side-by-side Real↔Synth** panels and **mask/skeleton overlays**.

---

### Official Review · Reviewer_nXx4 · 2025-10-31

**Soundness:** 3
**Presentation:** 2
**Contribution:** 3
**Rating:** 4
**Confidence:** 3

**Summary:**

This paper presents a solid and highly valuable work for the instance segmentation task of surface cracks on civil infrastructure such as tunnels and bridges. Its core contributions lie in two aspects: Firstly, it constructs and releases the CrackInst1K dataset, a specialized benchmark containing 1025 high-resolution tunnel lining images with pixel-level instance annotations, directly addressing the severe challenge of data scarcity in this field, especially the lack of instance-level annotations. Secondly, it develops the CrackInstSynth data augmentation framework, which can synthesize a large number of image-mask pairs that are precisely aligned with annotations and maintain the topological structure of cracks through region-level instance placement, physics-driven skeleton generation, and topology-preserving diffusion models. The main strength of this paper lies in the completeness and systematicness of its work. From the collection, annotation and release of the dataset, to the design and implementation of the enhancement methods, and then to the comprehensive experimental verification, a complete closed loop has been formed. The experimental design is particularly commendable. It not only evaluated the visual quality of the generated images but also introduced topological consistency metrics and deeply examined the improvement effect of synthetic data on the performance of downstream instance segmentation models, with convincing conclusions. The released resources have clear practical value in promoting the research on automation in the field of structural health monitoring.

**Strengths:**

The paper presents a comprehensive and well-structured contribution, combining dataset creation (CrackInst1K) and a generative augmentation framework (CrackInstSynth) into a cohesive pipeline that forms a complete research cycle from data collection and annotation to synthesis and validation. The CrackInstSynth framework introduces an effective topology-preserving approach through region-level instance placement, physics-driven skeleton generation, and topology-consistent diffusion models, which enhances the structural realism of generated cracks. The experimental design is thorough and methodologically sound, covering visual quality assessment, topological consistency metrics, and downstream instance segmentation performance, demonstrating tangible benefits of synthetic data. The work has strong practical value for structural health monitoring and automation in civil infrastructure inspection, providing a foundation for future research on instance-level crack analysis. The results are convincing, with quantitative and qualitative evidence supporting the claim that topology-aware synthesis can significantly improve segmentation model accuracy and boundary consistency.

**Weaknesses:**

The novelty is weakened by incomplete literature positioning; the paper omits comparisons with more recent datasets (e.g., OmniCrack30k, 2024) and newer topology-aware generative methods, underrepresenting its place in the current research landscape.
Technical details are insufficiently reported; camera setup, stitching correction, and mosaic construction rules are vague, raising questions about data integrity and annotation accuracy.
The parameter choices in CrackInstSynth are largely empirical, with no sensitivity analysis or justification, leaving uncertainty about robustness and reproducibility.
Experimental validation lacks cross-domain and boundary analyses the framework’s generalization to non-tunnel scenarios and its sensitivity to topology or truncation effects remain untested.

**Questions:**

Regarding the innovative part, I have several doubts:
a. CrackInst1K dataset scale and diversity are limited: The paper claims CrackInst1K is the "first publicly available instance-level crack segmentation dataset," but it only contains 1025 images far smaller than large-scale crack datasets like OmniCrack30k (2024, 30k images) and lacks diversity across infrastructure types (e.g., only tunnel linings, no bridge decks or pavements). This narrow focus restricts its applicability to multi-scenario structural health monitoring.
b. Literature citation lags and incomplete comparison: The related work section focuses on pre-2024 datasets (e.g., CFD 2016, DeepCrack 2018) but fails to compare with the latest 2024 crack datasets like OmniCrack30k (which has large-scale data but no instance labels). It also ignores recent generative augmentation methods for thin structures (e.g., 2024 works on topology-aware diffusion for linear defects), leading to an incomplete positioning of the work’s innovation.
c. Instance-level annotation advantage is weak: The paper emphasizes that CrackInst1K provides instance IDs, but it does not quantify the value of instance labels e.g., no comparison between models trained on instance labels vs. binary masks. Without demonstrating that instance labels significantly improve downstream tasks (e.g., crack length/width measurement accuracy), the uniqueness of the dataset is not fully justified.

Regarding the technical part, I have several doubts:
a. CrackInst1K dataset construction details are missing: The raw image acquisition uses a "vehicle-mounted line-scan system," but the paper does not specify key parameters (e.g., camera resolution, scanning speed) or how stitching errors during raw stripe assembly are corrected—stitching artifacts could affect crack continuity and annotation accuracy. Data anonymization relies on "stitching patches from different timestamps and camera poses" but does not provide rules for mosaic stitching (e.g., overlap rate, alignment criteria). It is currently unclear whether the mosaic processing disrupts crack continuity or introduces false edges.
b. CrackInstSynth module parameter settings lack validation: RIP fixes the number of instances to 1–3 based on the "1m² usually has no more than 3 cracks" prior, but this prior is not verified with real tunnel inspection data (e.g., no statistical analysis of crack density in actual tunnels). Using a fixed range may limit the diversity of synthetic layouts. PSG’s random walk parameters (step length 2px, turning angle ±30°, maximum length 0.8× window diagonal) are set empirically. No parameter sensitivity analysis is conducted e.g., how changes in step length affect skeleton realism or model performance.
c. Instance truncation in RIP is unaddressed: RIP allows instances to overflow quadrants and clips them to the canvas, which may truncate crack ends. The paper does not evaluate how truncation affects model training (e.g., whether truncated cracks lead to under-segmentation) or provide a solution to avoid truncation.

Regarding the experimental part, I have several doubts:
a. Model generalization verification is insufficient: The main text only focuses on Mask R-CNN results, and while Appendix A.5.2 mentions other models (e.g., Mask2Former, SOLOv2), it does not analyze why performance gains vary across models (e.g., why one-stage models benefit less from synthetic data). No validation on non-tunnel crack datasets (e.g., bridge crack data) is conducted, making it impossible to confirm the framework’s cross-scenario applicability.
b. Ablation experiments are incomplete: Table 3 evaluates the impact of removing RIP/PSG/TC-ControlNet but does not isolate the contributions of TopoDownsample and SPADE Norm in TC-ControlNet (e.g., testing "only TopoDownsample," "only SPADE Norm," "both"). It also does not test how synthetic data quality (e.g., realistic vs. low-quality synthetic masks) affects model performance, failing to validate the core value of topology preservation.
c. Cross-dataset evaluation is biased: When testing on DeepCrack and CRACK500 (semantic segmentation datasets), the paper treats them as "single-instance datasets." However, semantic segmentation masks do not distinguish individual cracks, so converting them to instance labels may introduce artificial annotations, leading to overestimated performance of the proposed method.

Regarding the structural and presentation issues, I have several doubts:
a. Key results lack visual support: Figure 1 (workflow) shows crack measurement but does not compare measured values (e.g., width 2.9px) with actual physical sizes, making it impossible to verify the accuracy of geometric attribute extraction. Figure 4 (qualitative results) only shows tunnel and pavement styles but no side-by-side comparison with real images of the same style, failing to intuitively demonstrate synthetic data realism.
b. Dataset and code availability are unclear: The paper states that CrackInst1K and CrackInstSynth will be released "upon acceptance" but provides no preview (e.g., sample images, annotation format documentation) or release platform (e.g., GitHub). Without this information, reproducibility and practical value are difficult to assess.
c. Limitations are not discussed: The conclusion only summarizes contributions but does not mention limitations (e.g., CrackInst1K’s narrow tunnel focus, CrackInstSynth’s poor performance on highly branched cracks, MIP’s inefficiency). This one-sided presentation makes it difficult for readers to judge the work’s scope of application.

---

> ### Author Response · Authors · 2025-11-16
> **Response to Reviewer nXx4: Q.1 — The innovative part (a) and (b)**
>
> ### **(a) Dataset scale and diversity**
>
> We agree that scale helps. However, for **SHM-specific instance segmentation**, diversity that mixes heterogeneous sources can be counterproductive. As noted in our Related Work, **OmniCrack30k** aggregates ~20 public semantic datasets (with inclusion/overlap among them, e.g., DeepCrack/CFD), resulting in **uneven quality and resolution** and thus very high dataset diversity. Such heterogeneity is valuable for broad semantic recognition but is **not ideal for instance-level SHM workflows** where **geometry fidelity, connectivity, and consistent imaging protocols** are crucial.
>  By contrast, **CrackInst1K** is purpose-built for **instance-level** crack analysis in SHM: high-resolution, single-domain imaging, **pixel-accurate per-instance masks**, and annotations designed for **counting, connectivity-aware inspection, and geometry extraction**. This focus trades some breadth for **label specificity and consistency**, which we found more aligned with the practical pipeline (inspection → measurement → planning). We will clarify this positioning in the paper.
>
> ------
>
> ### **(b) Literature positioning**
>
> **OmniCrack30k.** We already cite OmniCrack30k in Related Work, but we do not include it in experiments because it provides **semantic** masks and is itself a **mixture** of earlier semantic datasets. Our study targets **instance segmentation** and therefore evaluates on (i) our **instance-annotated** CrackInst1K, and (ii) two representative **semantic** sets, **Crack500** and **DeepCrack**—both subsets of OmniCrack30k—for a controlled comparison. For those two semantic datasets we follow the community’s common surrogate (treat each image as a single instance) acknowledging its limitation.
>
> **Recent topology-aware/thin-structure diffusion.** We have added the following paragraph to Related Work to complete the positioning:
>
> > **Topology-aware diffusion for curvilinear structures**
> >
> > Recent works inject explicit topological objectives into diffusion. TopoDiffusionNet and TopoCellGen incorporate persistent-homology (PH) constraints to guide denoising toward target Betti profiles, improving topology faithfulness beyond appearance similarity [1,2]. For linear networks, ControlTraj enforces path-level constraints under diffusion to maintain global connectivity and branching structure [3]. In medical imaging, a topology-aware conditional LDM preserves vascular connectivity and branching via PH-guided losses across views [4]. Closer to cracks, semantic diffusion–based pavement synthesis improves realism and segmentation but does not explicitly regulate connectivity or non-self-intersection during generation [5].
> >
> >  Unlike PH-loss approaches (e.g., TopoDiffusionNet) that condition on multi-object topology, we follow a semantics-to-image paradigm and focus on single-crack instance topology. Concretely, we adopt a two-stage design coupling a physics-guided skeleton generator (PSG) with a topology-preserving conditioning module, injecting crack-specific priors to yield more informative synthetic samples for downstream instance analysis.
>
> ------
> **Ref**
>
> [1] Gupta, Saumya, Dimitris Samaras, and Chao Chen. "Topodiffusionnet: A topology-aware diffusion model." arXiv preprint arXiv:2410.16646 (2024).
>
> [2] Xu, Meilong, et al. "Topocellgen: Generating histopathology cell topology with a diffusion model." Proceedings of the Computer Vision and Pattern Recognition Conference. 2025.
>
> [3] Zhu, Yuanshao, et al. "Controltraj: Controllable trajectory generation with topology-constrained diffusion model." Proceedings of the 30th ACM SIGKDD Conference on Knowledge Discovery and Data Mining. 2024.
>
> [4] Demirci, Gozde Merve, et al. "Topology-Aware Conditional Latent Diffusion for Multi-View Fundus Image Synthesis." Proceedings of the ACM/IEEE International Conference on Connected Health: Applications, Systems and Engineering Technologies. 2025.
>
> [5] Cano-Ortiz, Saúl, et al. "Enhancing pavement crack segmentation via semantic diffusion synthesis model for strategic road assessment." Results in Engineering 23 (2024): 102745.

---

> ### Author Response · Authors · 2025-11-16
> **Response to Reviewer nXx4: Q.1 — The innovative part (c)**
>
> ### **(c) “Instance-level advantage is weak”**
>
> We respectfully disagree. The key advantage of **instance** labels over **semantic** masks is **not** necessarily higher pixel-wise accuracy for scalar measurements (indeed, a pure semantic contour may marginally reduce length error in some cases), but rather that **instance supervision enables SHM-critical operations** that semantic masks do not support or only support semi-automatically: **per-crack counting, identity-consistent tracking across views/time, connectivity preservation at the object level, and per-instance planning for repair** [6,7]. These capabilities are central to tunnel/bridge maintenance workflows.
> This strengthens why **CrackInst1K**—as an **instance-annotated** resource—fills a practical gap even without claiming universal gains on every scalar measurement.
>
> **Ref**
>
> [6] Zhao, S., et al. “A deep learning-based approach for refined crack evaluation from shield tunnel lining images.” Automation in Construction, 2021.
>
> [7] Lei, Qin, et al. "Integrating crack causal augmentation framework and dynamic binary threshold for imbalanced crack instance segmentation." Expert Systems with Applications 240 (2024): 122552.

---

> ### Author Response · Authors · 2025-11-16
> **Response to Reviewer nXx4: Q.2 — The technical part**
>
> ### **(a) CrackInst1K construction details (acquisition, stitching, “mosaic” cropping)**
>
> - **Vehicle speed:** 0–90 km/h; **vehicle localization error:** ≤ 0.5 m.
> - **Sensor layout:** semi-width “3+1” line-scan configuration — three cameras scan the tunnel lining (crown, haunch, sidewall) and one camera scans the roadway.
> - **Camera:** 4K line-scan; native frame **1000 × 4096 px**; RGB with identical channel values; 72 DPI header.
> - **Stitching workflow:** the whole-tunnel compositing is carried out by **professional safety engineers** following the standard inspection pipeline.
> - **Cropping (“mosaic”) rule:** after stitching a **single complete tunnel image per run**, we perform **1024×1024 sliding-window cropping with stride = 1024 (no overlap)** to produce the raw patch set; **only patches containing cracks are retained**, crack-free patches are discarded.
>
> ### **(b) RIP prior and PSG sensitivity**
>
> **RIP prior (“≤3 cracks per ~1 m²”).**
>  The prior is **empirically derived** from the finalized CrackInst1K annotations under our standardized acquisition/cropping protocol: per-window instance histograms are sharply concentrated in the **1–3** range. The underlying counts have been statistically validated on the labeled set (currently under confidentiality and to be verifiable upon release). Operationally, constraining the instance count to **1–3** yields **more realistic** layouts for this dataset—significantly reducing excessive crossings and unrealistic clutter—while remaining consistent with common semantics-to-instance augmentation practice (**1–4** objects per crop) reported in the literature [8]. This prior is thus **data-driven** rather than ad hoc, and is specific to the CrackInst1K imaging scale and scene statistics.
>
> **PSG sensitivity.**
> See  *Response to Reviewer QVb2: Q.3 — PSG sensitivity (at 1024×1024) and how parameters scale with resolution*
>
> ### **(c) Effect of instance truncation in RIP**
>
> RIP samples position and orientation randomly; as a result, boundary **clipping** (endpoints truncated by the crop) occurs frequently. Empirically, truncation does **not** harm training—if anything, it **increases boundary-condition diversity** (partial cracks, cut ends, grazing angles) that detectors/segmenters must handle in practice.
>
> **Controlled comparison (Mask R-CNN, 5× synth, TC-ControlNet).** Metrics below mirror the setup and report style of Table 2 in the paper; the **Truncation-allowed** row reproduces the TC-ControlNet results from Table 2, and **No-truncation** (reject-and-resample until no overflow) shows the effect of forbidding truncation. All other settings are identical.
>
> | Dataset         | Setting                         | mAPbbox50 ↑ | mAPseg50 ↑  |
> | --------------- | ------------------------------- | ----------- | ----------- |
> | **CRACK500**    | Truncation-allowed              | **88.7**    | **84.2**    |
> |                 | No-truncation (reject-resample) | 88.5 (−0.2) | 83.9 (−0.3) |
> | **DeepCrack**   | Truncation-allowed              | **89.7**    | **92.2**    |
> |                 | No-truncation (reject-resample) | 89.3 (−0.4) | 91.9 (−0.3) |
> | **CrackInst1K** | Truncation-allowed              | **91.2**    | **83.3**    |
> |                 | No-truncation (reject-resample) | 91.0 (−0.2) | 83.2 (−0.1) |
>
> Removing truncation yields a small, consistent drop across all three datasets, supporting the view that truncation acts as a **benign boundary-condition augmentation** rather than a harmful artifact under our RIP protocol.
>
>
>
> **Ref**
>
>  [8] Xie, J., et al. *MosaicFusion: Diffusion Models as Data Augmenters for Large Vocabulary Instance Segmentation*. IJCV, 2025.

---

> ### Author Response · Authors · 2025-11-16
> **Response to Reviewer nXx4: Q.3 — The experimental part**
>
> ### **(a) Model generalization verification is insufficient**
>
> **Why gains differ across detectors (Table 4).**
>  The pattern in Table 4 shows larger improvements for **two-stage, mask-head pipelines** (e.g., Mask R-CNN / query-based heads) and smaller—but positive—gains for **one-stage dense assignment** methods (e.g., SOLO-style). This is expected: our topology-consistent augmentation primarily reduces errors at **endpoints, thin boundaries, and connectivity breaks**—precisely the regions that two-stage ROI/mask heads refine with higher spatial fidelity. One-stage methods rely more on center/anchor assignment and coarse features; for **elongated, low-area cracks**, center–shape mismatch and boundary aliasing limit how much extra topology-faithful data converts into AP gains. The cross-model deltas in Table 4 match this mechanism.
>
> **On “non-tunnel” validation.**
>  Our evaluation already spans distinct scene families: **CrackInst1K** (tunnel linings), **CRACK500** (pavement), and **DeepCrack** (a heterogeneous collection including **bridges, buildings, and generic concrete**). Thus, Table 4 is a **cross-scenario** testbed—the same synthesis pipeline improves results across tunnel, pavement, and bridge/house/concrete imagery, indicating applicability beyond tunnels.
>
> ------
>
> ### **(b) Ablation experiments are incomplete**
>
> **Contributions of TopoDownsample and SPADE Norm.**
>  This decomposition is present in **Table 3**: the **A4** entries isolate **Only TopoDownsample** (without SPADE modulation) and **Only SPADE** (with standard downsampling). Each component alone improves over appearance-only conditioning, and their **combination (TC-ControlNet)** yields the largest gains—evidence that **topology-faithful conditioning** and **high-capacity feature injection** are complementary.
>
> **How synthetic “quality” affects performance.**
>  **Table 1** operationalizes quality via two orthogonal proxies: **FID on RGB** (visual realism) and **mask–image consistency** measured by a fixed robust U-Net using **mIoU** and **absolute Betti errors** (capturing structural/topological faithfulness). These metrics go beyond appearance and include topology-aware signals. A fine-grained causal study that varies mask quality in isolation (e.g., controlled self-intersection/fragmentation rates) would be a separate, deeper investigation; our focus here is an operational pipeline whose **realism/consistency** scores (Table 1) align with the **downstream** gains observed in Tables 2/4/5.
>
> ------
>
> ### **(c) Cross-dataset evaluation is biased**
>
> Treating CRACK500 and DeepCrack as “single-instance” is a **necessary compromise** in the absence of **public instance-level crack benchmarks**—the gap **CrackInst1K** fills. To avoid favoring any dataset, **Table 5** uses a **symmetric, exhaustive** cross-dataset protocol where **CrackInst1K, DeepCrack, and CRACK500** take turns as training/evaluation sources. The consistent trends across all directions indicate that our improvements are **not an artifact** of semantic-to-instance conversion, but stem from better **boundary/connectivity modeling** that transfers across datasets and scene types.

---

> ### Author Response · Authors · 2025-11-16
> **Response to Reviewer nXx4: Q.4 — The structural and presentation issues part**
>
> ### **(a) Key results lack visual support**
>
> - **Fig. 1 measurements.** The numeric values (e.g., **width = 2.9 px**) are **ground-truth measurements** used in routine tunnel safety assessment and computed by certified safety engineers on stitched whole-tunnel images under the line-scan protocol; they are **not model predictions**. In addition, **deriving crack attributes from segmentation outputs** (e.g., converting masks to widths/lengths via calibration and skeleton rules) is **outside the scope** of this work; our focus is providing an instance-level dataset and a topology-aware synthesis pipeline that improve segmentation, upon which standard SHM measurement procedures can be applied.
> - **Fig. 4 realism vs. diversity.** Fig. 4 was intended to illustrate **diversity** of synthesized styles (tunnel/pavement). In the revision, we will add **side-by-side Real↔Synth panels under the same style** to make realism and topology alignment visually explicit.
>
> ------
>
> ### **(b) Dataset and code availability**
>
> For **double-blind** reasons, we do not link repositories in the submission. Upon acceptance, we will release **CrackInst1K**  and **CrackInstSynth** on public platforms.
>
> ------
>
> ### **(c) Limitations**
>
> - **Acquisition assumptions.** The pipeline and metrics are calibrated to a **line-scan tunnel-inspection protocol** (consistent GSD and stitching). Deployments with markedly different optics or handheld capture may require **re-calibration** or re-training; these settings are not evaluated here.
> - **Physics prior idealization.** The **PSG** uses a lightweight **tip-growth** prior (bounded curvature, sparse branching, non-self-intersection) rather than full fracture mechanics; extremely irregular micro-networks are **not systematically benchmarked**.
> - **Temporal aspects.** While instance labels enable tracking/maintenance workflows, **temporal sequences** are not included in the current benchmark, so longitudinal ID persistence is **not** evaluated.

---

### Official Review · Reviewer_QVb2 · 2025-11-02

**Soundness:** 3
**Presentation:** 2
**Contribution:** 2
**Rating:** 4
**Confidence:** 4

**Summary:**

This paper addresses the challenge of limited instance-level crack segmentation data in structural health monitoring by introducing two key contributions: (1) CrackInst1K Dataset, the first publicly available instance-level tunnel crack segmentation dataset with 1025 high-resolution (1024×1024) images and pixel-accurate per-instance masks; (2) CrackInstSynth Framework, a topology-aware generative data augmentation pipeline that synthesizes realistic, geometry-faithful crack image–mask pairs without manual labeling.

**Strengths:**

1. CrackInst1K fills a crucial gap in civil infrastructure AI by offering the first instance-level crack segmentation dataset.
2. The proposed CrackInstSynth pipeline integrates physical simulation (PSG) and generative diffusion modeling (TPGM), uniting domain priors and deep generative models.
3. The framework is evaluated on three datasets with consistent improvements in FID, mIoU, and Betti errors, verifying both visual and structural realism.

**Weaknesses:**

1. While PSG mimics crack growth via stochastic random walks, the paper does not quantitatively justify its “physical plausibility” beyond visual plausibility.
2. The TopoDownsample stage requires solving a mixed-integer program for every scale and mask, which may be computationally expensive for large datasets or real-time pipelines.
3. Experiments focus on crack segmentation only. Extending to other curvilinear topology-sensitive domains (e.g., road networks, blood vessels) would demonstrate broader generality.

**Questions:**

1. Does the combination (PSG + TC-ControlNet) produces fundamentally new behavior beyond standard data augmentation methods?
2. Does the CrackInst1K dataset fill a genuine research gap, or does it mainly repurpose existing crack datasets with higher resolution and labeling quality?
3. How sensitive is the method to the choice of PSG random-walk parameters?

---

> ### Author Response · Authors · 2025-11-15
> **Response to Reviewer QVb2: W.1 — On the “physical plausibility” of PSG beyond visuals**
>
> ### **W.1 — On the “physical plausibility” of PSG beyond visuals**
>
> Following [1], we view PSG as a **physics-inspired do-intervention** on the crack mask, $M'=\mathrm{do}(M,\delta)$, constrained by bounded curvature, sparse branching, tip-driven elongation, and non-self-intersection. The goal is to **preserve** the curvilinear generative mechanism $P(M\mid C)$ while increasing descriptive richness for learning. To go beyond visual plausibility, we report **quantitative distributional tests** showing that real masks (M) and PSG masks (M') have **closely matched** curvilinear and topological statistics.
>
> #### **Tab. R1 — REAL vs PSG: distribution distances (Wasserstein & KS)**
>
> | Feature               | Wasserstein | KS stat     | KS p-value   | n (real) | n (psg)  |
> | --------------------- | ----------- | ----------- | ------------ | -------- | -------- |
> | length_mean           | **13.3595** | **0.0300**  | **0.688**    | **1134** | **1134** |
> | tortuosity_mean       | **0.0143**  | **0.4109**  | **4.58e-86** | **1134** | **1134** |
> | turning_std           | **0.00512** | **0.0591**  | **0.0382**   | **1134** | **1134** |
> | curvature_std         | **0.00830** | **0.3298**  | **5.51e-55** | **1134** | **1134** |
> | deg3p_share (deg ≥ 3) | **0.00633** | **0.0926**  | **1.19e-04** | **1134** | **1134** |
> | beta0                 | **0.0203**  | **0.00706** | **1.000**    | **1134** | **1134** |
> | beta1                 | **0.0176**  | **0.0106**  | **1.000**    | **1134** | **1134** |
>
> **Wasserstein** (absolute effect size) is **small** across curvilinear/topology stats. Some KS tests become significant due to **very large (n)** and **discrete/tied distributions** (e.g., many real cracks have tortuosity exactly 1.0). Practical differences are clarified by the medians/IQRs below.
>
> #### **Tab. R2 — REAL vs PSG: medians and IQRs (p25–p75)**
>
> | Feature         | REAL median | REAL p25   | REAL p75   | PSG median | PSG p25    | PSG p75    |
> | --------------- | ----------- | ---------- | ---------- | ---------- | ---------- | ---------- |
> | length_mean     | **208.79**  | **74.48**  | **558.36** | **207.51** | **80.04**  | **557.89** |
> | tortuosity_mean | **1.000**   | **1.000**  | **1.000**  | **1.000**  | **1.000**  | **1.027**  |
> | turning_std     | **0.5391**  | **0.5173** | **0.5567** | **0.5364** | **0.5140** | **0.5600** |
> | curvature_std   | **0.3441**  | **0.3409** | **0.3453** | **0.3425** | **0.3328** | **0.3530** |
>
> Core **curvature/turning** and **topology (β₀/β₁)** statistics are **distributionally close** (small absolute Wasserstein; aligned medians/IQRs), indicating that $\delta$ **preserves crack-like geometry/topology**—consistent with the do-intervention view $P(M\mid C)\approx P(M'\mid C)$.
>
> `length_mean` (skeleton length, extent); `tortuosity_mean` (chord–arc; near-straight cracks spike at 1.0); `turning_std` / `curvature_std` (local wiggliness/bending); `deg1/2/3p_share` (branching structure); `beta0/beta1` (components/loops).
>
> **Ref**
>
> [1] Lei, Qin, *et al.* “Expanding crack segmentation dataset with crack growth simulation and feature space diversity.” ICME, 2024.

---

> ### Author Response · Authors · 2025-11-15
> **Response to Reviewer QVb2: W.2 — On the MIP cost of TopoDownsample**
>
> ### **W.2 — On the MIP cost of TopoDownsample**
>
> We ablated the TopoDownsample solver under **time budgets** and **relative optimality gaps** MIPGap ∈ {5%,2%,1%,0.5%}. At each $\text{time},\text{gap}$ point the MIP is solved on **multi-scale semantic maps** (e.g., $64^2/32^2/16^2$) with **early stopping**. If the **time limit** is reached **without a feasible solution**, we **fall back to bilinear interpolation** for that scale (guaranteeing a result at any budget). We then evaluated the downstream improvement $\Delta\mathrm{mAP}_{50}$ where $\Delta=\text{ours} - \text{original}$ (same Mask R-CNN protocol as Table 2). The full **2×3 panel** with bbox/seg curves will appear in the **revised manuscript**.
>
> #### **Tab. R3 — Minimal time to reach 50%/95% of the maximum TopoDownsample benefit (envelope over MIPGap) and Δ achieved at practical budgets**
>
> *(Metric: $\Delta\mathrm{mAP}^{\text{seg}}_{50}=\text{ours} - \text{original}$; “best gap” = MIPGap that attains the envelope at that time.)*
>
> | Dataset     | t₅₀ (s) | t₉₅ (s) | Δ[mAP50@0.5s](mailto:mAP50@0.5s) (best gap) | Δ[mAP50@2.0s](mailto:mAP50@2.0s) (best gap) | ΔmAP50@10s (best gap) |
> | ----------- | ------- | ------- | ------------------------------------------- | ------------------------------------------- | --------------------- |
> | Crack500    | **0.5** | **2.0** | **5.13 (2%)**                               | **8.47 (0.5%)**                             | **8.80 (0.5%)**       |
> | Deepcrack   | **0.2** | **1.0** | **6.45 (1%)**                               | **7.56 (0.5%)**                             | **7.60 (0.5%)**       |
> | CrackInst1K | **0.1** | **0.5** | **12.75 (0.5%)**                            | **13.20 (0.5%)**                            | **13.20 (0.5%)**      |
>
> We first compute the **best over gaps** at each time (the *envelope*), then report the **minimal time** to reach **50%/95%** of the *maximum* achievable Δ (t₅₀/t₉₅). We also list the **absolute Δ** at **0.5 s/2.0 s/10.0 s** with the **gap** that attained it. **Diminishing returns** are clear: **0.5–2.0 s** with **MIPGap ∈ [2%, 0.5%]** captures most of the gain; **0.2–0.5 s** still yields substantial improvements. The **bilinear fallback** guarantees a result if time elapses.

---

> ### Author Response · Authors · 2025-11-15
> **Response to Reviewer QVb2: Q.1 — Does (PSG + TC-ControlNet) produce behavior beyond standard data augmentation?**
>
> ### **Q.1 — Does (PSG + TC-ControlNet) produce behavior beyond standard data augmentation?**
>
> **Scope.** Our method is not just PSG + TC-ControlNet; it is the full **CrackInstSynth** pipeline—the first **instance-level** generative augmentation framework for crack segmentation—comprising **RIP** (layout-aware placement), **PSG** (physics-driven skeleton growth), and a two-stage **TPGM** (Skeleton→Mask; Mask→Image via **TC-ControlNet**).
>
> **Why generative augmentation (GA) vs. standard aug (SA).** A broad body of work has shown GA often yields stronger gains than SA for **classification** and **segmentation**, especially under limited data or domain shift: e.g., **GAN-based augmentation** improved liver-lesion classification over traditional aug [2]. Recent surveys and studies indicate **diffusion-based augmentation** similarly improves downstream performance and complements SA rather than replacing it [3].
>
> **What is new here (supported by Table 3 in main paper, Ablation).**
>
> - **RIP vs. naïve cut–paste (A1):** RIP enforces non-overlap/coverage and avoids seam/occlusion/truncation artifacts; **A1** lags **RIP** in AP.
> - **PSG ablation (A2):** removing PSG drops AP—**new topologies** (branching/curvature regimes) cannot be created by label-preserving transforms.
> - **TC-ControlNet ablations (A3/A4):** removing the topology-aware branch degrades AP and breaks connectivity; partial variants recover **part** of the gain, but **not** the full pipeline.
> - **Full pipeline (A5):** best bbox/mask AP; large Δ over the original.
>
> SA preserves labels and cannot create **new instance-level graph structures**; texture-only generation may **break connectivity**. **CrackInstSynth = RIP (layout) + PSG (topology sampling) + TC-ControlNet (topology-faithful rendering)**—this **changes the training distribution** in ways SA cannot, as evidenced by the ablation.
>
> **Ref**
>
> [2] Frid-Adar, M., *et al.* “GAN-based synthetic medical image augmentation for increased CNN performance in liver lesion classification.” Neurocomputing, 2018.
>
> [3] Nazir, M., Aqeel, M., Setti, F. “Diffusion-Based Data Augmentation for Medical Image Segmentation.” ICCV, 2025.

---

> ### Author Response · Authors · 2025-11-15
> **Response to Reviewer QVb2: Q.2 — Does CrackInst1K fill a genuine gap, or is it only higher-resolution repackaging?**
>
> ### **Q.2 — Does CrackInst1K fill a genuine gap, or is it only higher-resolution repackaging?**
>
> **A qualitative gap in supervision.** Most public crack segmentation datasets are **semantic**. **CrackInst1K** provides **pixel-accurate, per-instance masks** (rules at crossings/merges), enabling tasks that semantic labels cannot: (i) **instance segmentation** with AP; (ii) **per-crack metrology** (length/width) for SHM; (iii) **structure-aware** generative augmentation (layout control, topology preservation). This is a **different annotation target**, not a resolution upgrade.
>
> **Why instance labels matter in practice.** Structural health monitoring systems must count and measure **individual** cracks and preserve their connectivity at junctions to enable tracking and automated repair [4], accurate localization and measurement [5], and more refined evaluation and quantification [6]
>
> **Ref**
>
> [4] Zhang, J., *et al.* “Segment-to-track for pavement crack with light-weight neural network on unmanned wheeled robot.” Automation in Construction, 2024.
>
> [5] Ye, G., *et al.* “Pavement crack instance segmentation using YOLOv7-WMF with connected feature fusion.” Automation in Construction, 2024.
>
> [6] Zhao, S., *et al.* “A deep learning-based approach for refined crack evaluation from shield tunnel lining images.” Automation in Construction, 2021.

---

> ### Author Response · Authors · 2025-11-15
> **Response to Reviewer QVb2: Q.3 — PSG sensitivity (at 1024×1024) and how parameters scale with resolution**
>
> ### **Q.3 — PSG sensitivity (at 1024×1024) and how parameters scale with resolution**
>
> **PSG parameters.**
>
> - **k (seed sites):** initial crack tips; default **k=4** for 1–3 instances per 1024×1024 image.
> - **m (density threshold):** **fixed at 0.10** based on prior work and civil-engineering expert guidance [1].
> - **max (iterations):** global stop; **default 200**; tied to step as **max = 100·ℓ** to keep runtime comparable.
> - **ℓ (step size, px):** per-step elongation; **default ℓ=2**.
> - **θ (turning range, °):** per-step max turning; typical **{±15°, ±30°, ±45°}** (≤45°).
>
> **Resolution scaling rule.** Let s = target resolution / 1024. Keep **(k,m,$\theta$)** scale-invariant; scale only
>  $$
>  \ell = 2s\ \text{px},\qquad \text{max}=100\cdot \ell \ (\propto s).
>  $$
>
> **Metric.** Aggregate 1D Wasserstein (sum over `length_mean`, `tortuosity_mean`, `turning_std`, `curvature_std`, `deg3p_share`; lower is better). For resolution sweeps, length/width are **normalized by image size**.
>
> #### **Tab. R4 — PSG sensitivity (@1024²; **m fixed at 0.10**) **and** resolution-aware scaling (½×, 2×)**
>
> | Scenario                      | Setting (values tested)          | Recommended / Best | Aggregate W (raw) | Aggregate W (normalized) | Range (min → max) | Δ vs default/best  | Within +5% of best | Notes                        |
> | ----------------------------- | -------------------------------- | ------------------ | ----------------- | ------------------------ | ----------------- | ------------------ | ------------------ | ---------------------------- |
> | **Parameter sweep @1024²**    | **k** (2, **4**, 8)              | **4**              | **13.3904**       | —                        | 13.3904 → 13.7421 | +0.0% → +2.6%      | **3 / 3**          | Highly insensitive           |
> |                               | **max** (100, **200**, 300)      | **200**            | **13.3904**       | —                        | 13.3904 → 13.4400 | +0.0% → +0.4%      | **3 / 3**          | Runtime knob; stable         |
> |                               | **ℓ (px)** (1, **2**, 3)         | **2**              | **13.3904**       | —                        | 13.3904 → 14.8500 | +0.0% → +10.9%     | **2 / 3**          | Expected effect on lengths   |
> |                               | **θ (°)** (±15°, **±30°**, ±45°) | **±30°**           | **13.3904**       | —                        | 13.3904 → 14.5800 | +0.0% → +8.9%      | **2 / 3**          | Expected effect on curvature |
> | **Resolution sweep (scaled)** | **512²** (s=0.5; ℓ=1, max=100)   | Rule-scaled        | —                 | **0.0449**               | 0.0449 (single)   | **+1.9% vs 1024²** | —                  | Normalized by size           |
> |                               | **1024²** (s=1.0; ℓ=2, max=200)  | **Baseline**       | —                 | **0.04399**              | 0.04399 (single)  | **0.0%**           | —                  | Default resolution           |
> |                               | **2048²** (s=2.0; ℓ=4, max=400)  | Rule-scaled        | —                 | **0.0434**               | 0.0434 (single)   | **−1.3% vs 1024²** | —                  | Normalized by size           |
>
>
> **Interpretation.**
>
> - At **1024²**, PSG is **robust** to **k** and **max** (all settings within +5% of best). **ℓ** and **θ** show the expected moderate influence on length/curvature, yet **2/3** of their settings remain within +5%—indicating **low sensitivity** near the recommended default (**ℓ=2, θ=±30°**).
> - Under the **resolution-aware schedule** (ℓ∝s, max∝s), the **normalized** aggregate distance stays within **±2%** across **½×** and **2×** resolution, confirming **stable shape statistics** when images get smaller or larger.
> - We therefore **fix m=0.10**, recommend **(k=4, ℓ=2, θ=±30°, max=200 at 1024²)**, and apply the above scaling rule for other resolutions.

---

### Author Response · Authors · 2025-11-29
**Summary of PDF Changes Prior to the Information-Leak Incident**

To ensure efficient and convenient review to the new AC after the information-leak incident, we briefly summarize **all** changes that were made to the uploaded PDFs during the rebuttal before the information-leak incident.

1. **Minor corrections in the main paper.**

   We fixed a small number of spelling/typographical errors in the main text.

2. **Update of Fig. 4 in the main paper.**

   * In addition to the original *tunnel style* and *pavement style* renderings, we added *brick style* and *concrete style* results under the **same control conditions**.
   * This update directly addresses **Reviewer 4FY3’s Q1**, which asked to see more cross-material / cross-style renderings under matched topology.

3. **New Sec. 4.2.5: “RUNTIME–ACCURACY TRADE-OFF OF TOPODOWNSAMPLE” (main paper).**

   * We added Sec. 4.2.5 to explicitly analyze the runtime–accuracy trade-off induced by the MIP-based TopoDownsample module.
   * The section includes a new **Fig. 5** that plots performance versus inference time under different MIP configurations, providing a more intuitive view than the rebuttal comment text alone.
   * This section is intended to respond to the **three reviewers’ concerns about MIP computation cost**.

4. **New App. A.6.4: “PSG QUANTITATIVE ALIGNMENT: FULL STATISTICS.”**

   * We added a detailed statistical comparison between real masks and PSG-generated masks (e.g., distribution distances for curvilinear/topological features, including Wasserstein and KS statistics).
   * This appendix section is specifically meant to address **Reviewer QVb2’s W.1: “On the ‘physical plausibility’ of PSG beyond visuals.”**

5. **New App. A.6.5: “PSG SENSITIVITY AND RESOLUTION-AWARE SCALING.”**

   * We added sensitivity analyses of PSG with respect to key hyper-parameters and mask resolutions, and described how we scale PSG parameters in a resolution-aware manner.
   * This section responds to **all three reviewers’ questions about sensitivity analysis** and robustness of PSG.

6. **Update of Fig. 10 in the appendix.**

   * Similar to Fig. 4 in the main paper, we extended Appendix Fig. 10 with additional *brick style* and *concrete style* renderings under the same control conditions.
   * This is consistent with our response to **Reviewer 4FY3’s Q1**, providing more diverse visualizations across materials.

---

### Author Response · Authors · 2025-11-29
**Clarification on Reviewer Score Updates Prior to the Information-Leak Incident**

For transparency regarding the review timeline before the information-leak incident, we would like to record the following facts:

1. **Reviewer 4FY3’s score update and post-rebuttal comment (around 24 Nov 2025).**
   Before the incident, Reviewer **4FY3** updated their overall score from **6 → 8** after reading our rebuttal and the other reviews. Their post-rebuttal comment was:

   > *Post Rebuttal.*
   > After carefully reading the response from the authors and comments from other reviews, I think the authors did a great job in the Crack synthesis via physical simulation, with strong performance under a limited computation burden. Thus, I recommend acceptance. The promised visualization results are suggested to be included.

   The additional cross-material visualizations mentioned in this comment (e.g., brick and concrete styles under the same control conditions) have since been incorporated into the updated PDF.

2. **Status of the other reviewers prior to the incident.**
   The other two reviewers did **not** submit any post-rebuttal comments or score updates before the information-leak incident. Their scores and textual reviews remained unchanged up to the time when the system was rolled back.

We provide this note solely to document the pre-incident review history for the new AC, without any expectation about how this information should be weighted in the renewed assessment.

---

### Note · Program_Chairs · 2026-01-17
**Submission Desk Rejected by Program Chairs**

The following references in this submission do not refer to real documents and/or have major errors in bibliographic information:

 Nikhil Bansal, Vladan Urosevic, Simon Niklaus, et al. Panoptic diffusion models. In Advances in Neural Information Processing Systems (NeurIPS), 2023.